# VisRAG: Vision-based Retrieval-augmented Generation on Multi-modality Documents

**Shi Yu**[1*]**, Chaoyue Tang**[2*]**, Bokai Xu**[2*]**, Junbo Cui**[2*]**, Junhao Ran**[3]**, Yukun Yan**[1†]**,**
**Zhenghao Liu**[4]**, Shuo Wang**[1]**, Xu Han**[1]**, Zhiyuan Liu**[1†]**, Maosong Sun**[1]
[1]Department of Computer Science and Technology, Tsinghua University
[2]ModelBest Inc. [3]Rice University [4]Northeastern University
yus21@mails.tsinghua.edu.cn

## Abstract

Retrieval-augmented generation (RAG) is an effective technique that enables large language models (LLMs) to utilize external knowledge sources for generation. However, current RAG systems are solely based on text, rendering it impossible to utilize vision information like layout and images that play crucial roles in real-world multi-modality documents. In this paper, we introduce VisRAG, which tackles this issue by establishing a vision-language model (VLM)-based RAG pipeline. In this pipeline, instead of first parsing the document to obtain text, the document is directly embedded using a VLM as an image and then retrieved to enhance the generation of a VLM. Compared to traditional text-based RAG, VisRAG maximizes the retention and utilization of the data information in the original documents, eliminating the information loss introduced during the parsing process. We collect both open-source and synthetic data to train the retriever in VisRAG and explore a variety of generation methods. Experiments demonstrate that VisRAG outperforms traditional RAG in both the retrieval and generation stages, achieving a 20–40% end-to-end performance gain over traditional text-based RAG pipeline. Further analysis reveals that VisRAG is efficient in utilizing training data and demonstrates strong generalization capability, positioning it as a promising solution for RAG on multi-modality documents. Our code and data are available at https://github.com/openbmb/visrag.

## 1 Introduction

Trained on massive data, large language models (LLMs) have shown strong abilities in common NLP tasks using their parametric knowledge (Wei et al., 2022; Zhao et al., 2023; Achiam et al., 2023). However, the issue of hallucination (Ji et al., 2023; Bang et al., 2023) and the challenge of updating the parametric knowledge limit their real-world application in specific domains. Retrieval-augmented generation (RAG) alleviates this problem by supplying the LLM with information retrieved from a custom outer knowledge base (Guu et al., 2020; Lewis et al., 2020; Yu et al., 2023). Open-source RAG frameworks like llamaindex (Liu, 2022) have been developed to facilitate the research and deployment of RAG.

Typical retrieval-augmented generation (RAG) pipelines are *text-based*, operating on segmented texts as retrieval units (Yu et al., 2023; Asai et al., 2024; Yan et al., 2024), which we refer to as TextRAG. In real-world scenarios, knowledge is often presented in multi-modality documents such as textbooks and manuals, which may have texts and figures intersected together. To acquire texts from such data sources, a *parsing* stage is required, which typically involves a cascade of processes, including layout recognition, optical character recognition (OCR), and post-processing steps like text joining (Zhang et al., 2024; Liu, 2022). While effective in most scenarios, the parsing process inevitably introduces errors, which can negatively impact the retrieval and generation phases. More-over, TextRAG utilizes only textual information, overlooking potential information present in other modalities like images. Although research has been conducted on image retrieval and multi-modal

---

*Equal contribution.
†Corresponding authors.

RAG, these approaches primarily focus on predefined scenarios wherein images and descriptive texts are properly extracted and paired (Wei et al., 2023; Sharifymoghaddam et al., 2024; Zhou et al., 2024), differing from real-world scenarios where texts and images (including figures) are often interleaved within a single document page.

The recent development of vision-language models (VLMs) has introduced a promising approach to understanding complex visual cues in images and documents (OpenBMB, 2024b; Wang et al., 2024). By integrating a language model with a vision encoder, VLMs demonstrate superior abilities in applications such as describing pictures (Alayrac et al., 2022), explaining figures (Bavishi et al., 2023), and transcribing (printed and handwritten) text from document images (Laurençon et al., 2024). Given the robust capabilities of VLMs in capturing multi-modal information present in images, an intriguing question arises: can the basic language model in the retrieval and generation components of TextRAG be substituted with a VLM, thus the parsing stage is bypassed and all the information of the document is preserved?

In this paper, we present **Vis**ion-based **R**etrieval-**a**ugmented **G**eneration (VisRAG), to study the feasibility of building a pure-vision RAG pipeline using VLMs. VisRAG is built with a VLM-based retriever VisRAG-Ret and generator VisRAG-Gen. Inherited the bi-encoder of text-based dense retriever (Karpukhin et al., 2020), VisRAG-Ret maps the query and the document into an embedding space, but utilizing the document's image directly instead of relying on extracted textual content. The embedding is obtained by applying weighted mean pooling on the final hidden states of the input text or vision tokens. After retrieving top-$k$ document images, VisRAG processes these images to generate the answer. While it is straightforward to use a VLM that supports multi-image input for generation, for VLMs that can only accept one single image, we propose page concatenation and weighted selection techniques to enable the handling of multiple documents. Throughout the process, VisRAG preserves all information in its original visual format, thereby preventing the potential information loss or distortion that might occur in traditional RAG pipelines.

To evaluate VisRAG on real-world multi-modal documents, we construct datasets from open-source visual question answering (VQA) datasets and synthetic query-document pairs derived from web-crawled PDFs. In terms of retrieval, VisRAG-Ret outperforms state-of-the-art text- and vision-centric retrievers and achieves better results than solely relying on its constituent vision encoder or language model under identical training conditions. For generation, VisRAG-Gen surpasses traditional text-based generators with open-source VLMs. With VLMs capable of handling multiple images, VisRAG shows increasing performance gains with more retrieved documents, indicating the potential for multi-page reasoning. As depicted in Figure 1, in a direct comparison of pipeline performances, VisRAG achieves a 40% relative improvement over TextRAG using MiniCPM-V 2.6 (OpenBMB, 2024b) as the generator and a 20% relative improvement with GPT-4o (OpenAI, 2024) as the generator, attributed to the cascade effect. Further analysis reveals that VisRAG possesses

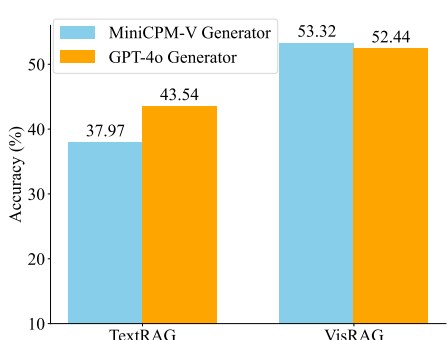

Figure 1: TextRAG vs. VisRAG on final generation accuracy. In TextRAG, parsed text serves as the basis for both retrieval and generation processes. In contrast, VisRAG leverages the original document image directly by using a VLM-based retriever and generator. Details can be found in Sec. 5.1.

better training data efficiency and generalization ability than baseline models, and demonstrates robustness across both text-centric and vision-centric documents. VisRAG shows great promise in replacing TextRAG as the next-generation standard for RAG pipelines.

## 2 RELATED WORK

**Retrieval-augmented Generation (RAG).** RAG enhances large language models (LLMs) by incorporating retrieved information from external knowledge bases, which assists in addressing knowledge-intensive tasks (Guu et al., 2020), reducing hallucinations (Semnani et al., 2023), and

acquiring new knowledge (Vu et al., 2023). An RAG pipeline typically comprises a text-based retriever that fetches relevant information from the knowledge base given the user query, and an LLM-based generator that reads the query along with the retrieved information to generate an answer (Shi et al., 2024b; Yu et al., 2023). Prior research on RAG primarily focuses on: a) improving the retriever, which is typically a text encoder producing text embeddings, through generator feedback (Yu et al., 2023; Shi et al., 2024b); b) enhancing the generator via supervised fine-tuning (Lin et al., 2024; Xu et al., 2024a), in-context pre-training (Shi et al., 2024a), or advanced prompting (Xu et al., 2024c); and c) developing advanced RAG pipelines to handle long-form or multi-hop question answering (Jiang et al., 2023; Asai et al., 2024). However, research on RAG has predominantly targeted cleaned text corpora like Wikipedia from an academic standpoint. Building effective RAG pipelines for real-world, multi-modal documents remains a challenge.

**Vision-language Models.** Recent advancements in vision-language models (VLMs) have greatly improved fine-grained multi-modal understanding. Since CLIP (Radford et al., 2021) pioneered contrastive visual-text alignment, models like Flamingo (Alayrac et al., 2022), LLaVA (Liu et al., 2023b), and BLIP (Li et al., 2022) have expanded LLMs to process visual inputs by connecting languages models with a CLIP-style vision encoder. Research has then shifted towards more advanced multi-task and multi-stage pre-training paradigms, enabling models to generalize across a wide range of vision-language tasks (Liu et al., 2024a; Bai et al., 2023; Wang et al., 2023; Dai et al., 2023). This is followed by notable advancements in high-resolution visual understanding (Xu et al., 2024b; Bavishi et al., 2023; Lin et al., 2023) and OCR capabilities (Kim et al., 2022; Lee et al., 2023; Hong et al., 2024; Chen et al., 2024b). Specifically, VLMs like the DocOwl series (Ye et al., 2023a; Hu et al., 2024b;a), UReader (Ye et al., 2023b), and TextMonkey (Liu et al., 2024b) are purpose-built to tackle OCR-free document understanding. More recently, breakthroughs have been made in multi-image understanding (Li et al., 2024a; Wang et al., 2024). Recent open-source VLMs like the MiniCPM-V (Yao et al., 2024) and Qwen2-VL (Wang et al., 2024) series combine the merits of recent techniques, achieving state-of-the-art performance. Those features of VLMs provide a foundation for our vision-based RAG pipeline, which requires multi-modal document understanding.

**Multi-modality Retrieval and RAG.** Multi-modal retrieval encompasses a wide range of tasks, such as retrieving a matching image given the text (Han et al., 2017), retrieving a text-image pair to answer a question (Chang et al., 2022), and retrieving texts that answer the given query about a provided image (Hu et al., 2023a; Luo et al., 2023), etc. Wei et al. (2023) propose UniIR, a universal multi-modal retrieval model capable of addressing the aforementioned multiple tasks. The retrieved information is then employed for incorporating knowledge (Hu et al., 2023b; Luo et al., 2021) or in-context learning (Tan et al., 2024; Liu et al., 2023a), with the aim of generating answers or images (Sharifymoghaddam et al., 2024). Prior research mentioned above is conducted on academic datasets, where texts and images are meticulously extracted from raw data and paired (e.g., images with their captions), to make it feasible to do separate encoding of data in different modalities. This hinders their applicability in real-world RAG scenarios, as real-world multi-modal documents are often presented in mixed modalities, and information may be distributed across various combinations of modalities. Concurrent works DSE (Ma et al., 2024) and ColPali (Faysse et al., 2024) address this issue by directly encoding the image of a document for retrieval. However, as these studies focus on retrieval, they lack a comprehensive comparison of their approaches with text-based retrieval in both in-domain and out-of-domain settings, and do not conduct an end-to-end RAG evaluation.

## 3 METHODOLOGY

In this section, we first recap the typical RAG pipeline (Sec. 3.1), then present our VisRAG framework (Sec. 3.2) and the construction of our training and evaluation data (Sec. 3.3).

### 3.1 PRELIMINARY: RETRIEVAL-AUGMENTED GENERATION

A typical retrieval-augmented generation (RAG) pipeline consists of a retriever and a generator, both built on large language models (LLMs)[1]. This pipeline operates on a knowledge corpus $\mathcal{D}$,

---

[1]In many cases, the retriever uses language models smaller than 1B parameters, which may not be considered "large", but we use the term LLM for simplicity.

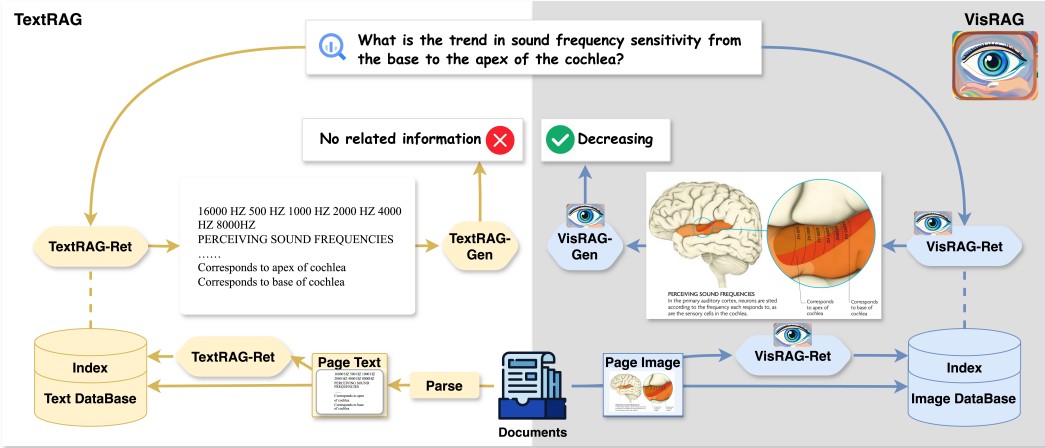

Figure 2: TextRAG (left) vs. VisRAG (right). Traditional text-based RAG (TextRAG) relies on parsed texts for retrieval and generation, losing visual information in multi-modal documents. Our vision-based RAG (VisRAG) employs a VLM-based retriever and generator to directly process the document page's image, thereby preserving all information in the original page.

which is processed into units for retrieval and generation, denoted as $\mathcal{D} = \{d_1, \ldots, d_n\}$, where $n$ is the number of retrieval units. Given a text query $q$ and the retrieval corpus $\mathcal{D}$, the retriever functions as $\mathcal{R} : (q, \mathcal{D}) \to \mathcal{D}_{\mathcal{R}}$, taking $q$ and $\mathcal{D}$ as inputs and producing a candidate set $\mathcal{D}_{\mathcal{R}} \subset \mathcal{D}$. To enable efficient search, the units in the knowledge corpus $\mathcal{D}$ are pre-encoded into embeddings. During RAG pipeline inference, approximate nearest neighbor (ANN) search is applied to retrieve $\mathcal{D}_{\mathcal{R}}$, which serves as the knowledge source for generation. The generation process can be defined as a function $\mathcal{G} : (q, \mathcal{D}_{\mathcal{R}}) \to a$, where $a$ represents the answer and $\mathcal{G}$ denotes the LLM generator. This is achieved by prompting the LLM with the query and the retrieved units $\mathcal{D}_{\mathcal{R}}$ to generate an answer.

As shown in Figure 2 (left), traditional RAG frameworks (TextRAG) typically utilize text-based units for retrieval and generation. However, in real-world scenarios, data often appear in complex, multi-modal documents, requiring an additional parsing step to obtain text. In this paper, we propose to use the *page* as the fundamental unit for retrieval and generation, which is directly processed by vision language models (VLMs) as an image without further processing during retrieval and generation. In subsequent sections, we use the terms "page" and "document" interchangeably.

### 3.2 VISRAG: VISION-BASED RETRIEVAL-AUGMENTED GENERATION

In this section, we present **Vis**ion-based **R**etrieval-**a**ugmented **G**eneration (VisRAG), as shown in Figure 2 (right). In contrast to traditional RAG frameworks which use text segments for both retrieval and generation, VisRAG leverages the image of the document to preserve all information.

#### 3.2.1 RETRIEVAL

The first stage of VisRAG, VisRAG-Ret, aims to retrieve a set of pages from the corpus $\mathcal{D}$ given $q$. We follow the dual-encoder paradigm in text-based dense retrieval models (Karpukhin et al., 2020) but employ a VLM rather than an LLM to encode the query and page. Specifically, the query and page are encoded separately as text and image in the VLM, producing in a sequence of hidden states. To derive the final embedding, and given that we use generative VLMs with causual attention, we adopt the position-weighted mean pooling over the last-layer VLM hidden states (Muennighoff, 2022), giving higher weights to later tokens:

$$\mathbf{v} = \sum_{i=1}^{S} w_i \mathbf{h}_i, \tag{1}$$

where $\mathbf{h}_i$ is the $i$-th hidden state, $S$ is the sequence length, $w_i = \frac{i}{\sum_{j=1}^{S} j}$ is the $i$-th weight, and $\mathbf{v}$ is the query or page embedding. The similarity score is calculated by the cosine similarity of the query

and page embedding. VisRAG-Ret is optimized using the InfoNCE loss:

$$l(q, d^+, D^-) = -\log \frac{\exp(s(q, d^+)/\tau)}{\exp(s(q, d^+)/\tau) + \sum_{d^- \in D^-} \exp(s(q, d^-)/\tau)}, \tag{2}$$

where $d^+$, $D^-$ are positive document and the negative document set of $q$, respectively, $s(q, d)$ is the similarity score between $q$ and $d$, and $\tau$ is the temperature.

### 3.2.2 GENERATION

The second stage of VisRAG, VisRAG-Gen, focuses on generating the answer according to the user query and retrieved pages using a VLM. We propose the following mechanisms to enable VisRAG-Gen to handle multiple retrieved pages in $\mathcal{D}_\mathcal{R}$ for generation. The prompts used for generation is presented in Appendix E.

**Page Concatenation.** A straightforward approach is to concatenate all pages in $\mathcal{D}_\mathcal{R}$ into a single image to accommodate most VLMs that are trained to accept a single image. Formally,

$$a \leftarrow \text{VLM-Single}(q, \text{Concat}(\{d | d \in \mathcal{D}_\mathcal{R}\})), \tag{3}$$

where VLM-Single is a VLM that accepts a single image with text prompt and Concat is the image concatenation operation. In this paper, we experiment with horizontal concatenation.

**Weighted Selection.** Another approach is to ask the VLM to generate an answer for every page from top-$k$, and select a final one with the highest confidence (Lewis et al., 2020; Shi et al., 2024b). The final confidence is defined as the weighted generation probability of the answer:

$$P(a|q, \mathcal{D}_\mathcal{R}) = P(a|q, d) \cdot \lambda(q, d), \tag{4}$$

where $P(a|d, q)$ is calculated as the reciprocal of the perplexity of generating the answer $a$ conditioned on the single document $d$, and $\lambda(d, q)$ is the normalized retrieval score:

$$\lambda(q, d) = \frac{e^{s(q,d)}}{\sum_{d' \in \mathcal{D}_\mathcal{R}} e^{s(q,d')}}. \tag{5}$$

**VLMs Accepting Multiple Images.** Some recent VLMs like MiniCPM-V 2.6 (OpenBMB, 2024b) and Qwen-VL 2 (Wang et al., 2024) are designed and trained to accept multiple images as input to perform cross-image reasoning. This capability may be useful for the generation as the required information could be located on a single page from the retrieved document set $\mathcal{D}_\mathcal{R}$ for single-hop questions or spread across multiple pages for multi-hop questions. Formally, we have

$$a \leftarrow \text{VLM-Multi}(q, \{d | d \in \mathcal{D}_\mathcal{R}\}), \tag{6}$$

where VLM-Multi is the VLM that accepts multiple images with text prompt.

### 3.3 DATA CONSTRUCTION

To effectively build and evaluate RAG pipelines on multi-modal documents, we construct our datasets using a combination of visual question answering (VQA) datasets and synthetic data. The statistics of our constructed dataset are provided in Table 1.

**Data Sources.** We collect question-document pairs from a series of VQA datasets, targeting different document types: MP-DocVQA (Tito et al., 2023) for industrial documents, ArXivQA (Li et al., 2024b), ChartQA (Masry et al., 2022), InfographicsVQA (Mathew et al., 2022), and PlotQA (Methani et al., 2020) for various figure types, and SlideVQA (Tanaka et al., 2023) for presentation slides. All datasets feature questions that can be answered using a single document (page), except for SlideVQA, which includes multi-hop questions requiring information from multiple pages. We follow the original datasets' train-test splits, except for MP-DocVQA and InfographicsVQA, where the validation split serves as our evaluation set. Additionally, we enhance our training set by collecting openly available PDFs from online sources and generating queries using GPT-4o (OpenAI, 2024), with details presented in Appendix A.1. We assemble the retrieval corpus by gathering the document associated with each query from the training and evaluation sets.

Table 1: Dataset statistics. We collect data from visual question answering (VQA) datasets for training and evaluation and synthetic additional query-document pairs for training. We apply filtering on VQA datasets to remove context-dependent queries that are not suitable for retrieval.

| Source | Document Type | Train | Evaluation | | |
| --- | --- | --- | --- | --- | --- |
| | | # Q-D Pairs | # Q (% Preserved) | # D | # Pos. D per Q |
| ArXivQA (2024b) | Arxiv Figures | 25,856 | 816 (8%) | 8,066 | 1.00 |
| ChartQA (2022) | Charts | 4,224 | 63 (5%) | 500 | 1.00 |
| MP-DocVQA (2023) | Industrial Documents | 10,624 | 591 (11%) | 741 | 1.00 |
| InfoVQA (2022) | Infographics | 17,664 | 718 (26%) | 459 | 1.00 |
| PlotQA (2020) | Scientific Plots | 56,192 | 863 (4%) | 9,593 | 1.00 |
| SlideVQA (2023) | Slide Decks | 8,192 | 556 (25%) | 1,284 | 1.26 |
| Synthetic | Various | 239,358 | - | - | - |

**Query Filtering.** Some queries extracted from VQA datasets are *context-dependent*, which lack specificity to a certain entity. For instance, the response to "Where was the conference held?" varies based on the contextual document. Using such context-dependent queries in open retrieval tasks is ineffective because they lack strong document specificity. To address this, we implement an additional filtering stage to remove these context-dependent questions, where we prompt GPT-4o (OpenAI, 2024) with human-annotated in-context samples to generate the classification label. Table 1 shows a substantial reduction in context-dependent questions across evaluation sets. The details of filtering are presented in Appendix A.2.

**Evaluation Metrics.** We report the retrieval and generation performance on the evaluation sets of the datasets sourced from VQA datasets. For retrieval, we use MRR@10 and Recall@10 as the metrics. For generation, consistent with methods applied to the source datasets, we report the answer accuracy, employing a relaxed exact match metric which allows a 5% error margin for numeric responses (Masry et al., 2022; Methani et al., 2020).

## 4 EXPERIMENTAL METHODOLOGY

In this section, we introduce our setup for experiments. Descriptions of the LLMs/VLMs used in our experiments can be found in Appendix C.

**Document Parsing.** To evaluate the performance of VisRAG against TextRAG, we introduce two text extraction methods. The first, "(OCR)", employs a pipeline that uses PPOCR (Du et al., 2020) to detect text regions and then merges nearby boxes to reduce fragmentation. The second, "(Captioner)", is a model-based approach that directly extracts text from document images using MiniCPM-V 2.0 (OpenBMB, 2024a; Yao et al., 2024) fine-tuned on paired (document image, extracted text) data. More details are provided in Appendix B.

**Retrieval Experiments.** VisRAG-Ret is a document embedding model built on MiniCPM-V 2.0, a vision-language model that integrates SigLIP (Zhai et al., 2023) as the vision encoder and MiniCPM (Hu et al., 2024c) as the language model. To ensure fair comparisons, we organize experiments into three settings: off-the-shelf, out-of-domain, and in-domain, as depicted below. We report VisRAG-Ret's performance in both out-of-domain and in-domain settings.

- Off-the-shelf: We directly evaluate popular text and image retrieval models on extracted texts, including BM25 (OCR), a lexical model; bge-large-en-v1.5 (Xiao et al., 2023) (OCR) and NV-Embed-v2 (Lee et al., 2024) (OCR), state-of-the-art text embedding models with sizes 335M and 7.85B, respectively; and SigLIP, a CLIP-style (Radford et al., 2021) vision model serving as the encoder for MiniCPM-V series.

- Out-of-domain: Out-of-domain models are trained solely on synthetic data and evaluated on the VQA datasets without in-domain supervision. These models include MiniCPM (OCR), MiniCPM (Captioner), and SigLIP. MiniCPM (OCR) and (Captioner) are MiniCPM-based text embedding models trained and evaluated on extracted text.

Table 2: Overall retrieval performance in MRR@10. The best retrieval performance in each group is marked in **bold**, and the second best performance is underlined. We train ColPali (Faysse et al., 2024) on our dataset. Corresponding Recall@10 performance can be found in Table 6.

| Model | # Para. | ArxivQA | ChartQA | DocVQA | InfoVQA | PlotQA | SlideVQA | Average |
|---|---|---|---|---|---|---|---|---|
| (a) Off-the-shelf Models | | | | | | | | |
| BM25 (OCR) | n.a. | 43.65 | 61.47 | 75.27 | 66.94 | 57.28 | 86.78 | 65.23 |
| bge-large (2023) (OCR) | 335M | 39.29 | 59.64 | 50.76 | 72.38 | 51.33 | 81.38 | 59.13 |
| NV-Embed-v2 (2024) (OCR) | 7.85B | **59.39** | **80.47** | **75.46** | **84.24** | **59.36** | **92.49** | **75.24** |
| SigLIP (2023) | 883M | 31.39 | 64.71 | 46.56 | 62.85 | 30.23 | 75.14 | 51.81 |
| (b) Out-of-domain: *Models Fine-tuned on Synthetic Data* | | | | | | | | |
| MiniCPM (OCR) | 2.72B | 47.96 | 61.64 | 67.04 | 79.36 | 36.04 | 87.93 | 63.33 |
| MiniCPM (Captioner) | 2.72B | 42.07 | **71.84** | 64.48 | 76.10 | 29.76 | 81.01 | 60.88 |
| SigLIP (2023) | 883M | 46.81 | 68.40 | 57.61 | 67.12 | 31.92 | 85.14 | 59.50 |
| VisRAG-Ret | 3.43B | **69.17** | 66.37 | **73.06** | **84.65** | **45.57** | **90.09** | **71.49** |
| (c) In-domain: *Models Fine-tuned on Synthetic and In-domain data* | | | | | | | | |
| MiniCPM (OCR) | 2.72B | 58.43 | 77.74 | 72.54 | 83.45 | **64.78** | 91.74 | 74.78 |
| MiniCPM (Captioner) | 2.72B | 56.15 | 74.06 | 67.57 | 81.22 | 55.43 | 84.27 | 69.78 |
| SigLIP (2023) | 883M | 59.16 | **81.34** | 64.60 | 74.59 | 61.32 | 89.08 | 71.68 |
| ColPali (2024) | 2.92B | 72.50 | 73.49 | **82.79** | 81.15 | 55.32 | **93.99** | 76.54 |
| VisRAG-Ret | 3.43B | **75.11** | 76.63 | 75.37 | **86.37** | 62.14 | 91.85 | **77.91** |

- In-domain: Models in this category are trained on the blend of the VQA training data and synthetic data. We evaluate the same set of models as in the out-of-domain setting to show model performance when supervised labels are available. We also report the performance of ColPali (Faysse et al., 2024) on our evaluation data. ColPali is a page embedding model that encodes a screenshot of a page into multiple vectors. We train ColPali on our dataset using the official code and hyper-parameters provided in its paper.

**Generation Experiments.** To evaluate generation performance, we fix the retrieval model to VisRAG-Ret and report the performance of various generation models and methods. For VisRAG-Gen, we compare the performance of the single-image VLM MiniCPM-V 2.0, which only accepts a single image, against the multi-image VLM MiniCPM-V 2.6 (OpenBMB, 2024b; Yao et al., 2024) and GPT-4o (OpenAI, 2024). MiniCPM-V 2.6 is an upgrade of MiniCPM-V 2.0, incorporating Qwen2-7B (Yang et al., 2024) as the language model and supporting multi-image input. We evaluate the performance of page concatenation and weighted selection on the single-image VLM. Additionally, we report the performance of text-based generation baselines, including MiniCPM (OCR) and GPT-4o (OCR), where only extracted texts are used for generation. For all experiments, we report results using the top-1, top-2, and top-3 retrieved documents, as well as an "Oracle" condition where the model is provided with only the positive document(s) to show the performance upper bound.

**Implementation Details.** VisRAG-Ret is fine-tuned using in-batch negatives (Karpukhin et al., 2020) for one epoch with a batch size of 128 on 8 NVIDIA A100 80GB GPUs. The temperature parameter in Equation 2 is set to 0.02. Baseline retrievers are fine-tuned with the same hyper-parameters, and textual baselines utilize extracted text data as document-side input. The generation part does not use any fine-tuning; we directly use off-the-shelf LLMs/VLMs for generation.

## 5 EVALUATION RESULTS

In this section, we first present the overall performance of VisRAG (Sec. 5.1), followed by analyses of training data efficiency (Sec. 5.2) and performance on different subsets (Sec. 5.3).

### 5.1 OVERALL PERFORMANCE

**Retrieval Performance.** In this experiment, we compare VisRAG-Ret with (a) off-the-shelf models, and trained baselines in (b) out-of-domain setting where we only leverage synthetic data, and in (c) in-domain setting where we leverage both in-domain and synthetic training data.

As shown in Table 2(a)(b), VisRAG-Ret, trained on out-of-domain data, significantly outperforms both off-the-shelf models BM25 and bge-large, and achieves 95% of the performance of NV-Embed-v2, a state-of-the-art text retrieval model with 7.85B parameters. Note that bge-large and NV-

Table 3: Overall generation performance in accuracy (%). All models and methods utilize the same retriever, VisRAG-Ret. Performance relative to Oracle is colored in blue.

| Model / Method | Input | ArxivQA | ChartQA | DocVQA | InfoVQA | PlotQA | SlideVQA | Average |
|---|---|---|---|---|---|---|---|---|
| (a) TextRAG-Gen: *Text-based Generation* | | | | | | | | |
| MiniCPM (OCR) | top-1 | 43.38 (96.2%) | 25.40 (72.7%) | 31.47 (75.9%) | 20.19 (92.9%) | 16.34 (94.0%) | 29.32 (94.8%) | 27.68 (87.8%) |
| | top-2 | 42.16 (93.5%) | 23.81 (68.2%) | 33.67 (81.2%) | 20.19 (92.9%) | 14.14 (81.3%) | 30.40 (98.3%) | 27.39 (85.9%) |
| | top-3 | 44.12 (97.8%) | 20.63 (59.1%) | 31.81 (76.7%) | 18.25 (84.0%) | 16.34 (94.0%) | 29.14 (94.2%) | 26.71 (84.3%) |
| | Oracle | 45.10 (100%) | 34.92 (100%) | 41.46 (100%) | 21.73 (100%) | 17.38 (100%) | 30.94 (100%) | 31.92 (100%) |
| GPT-4o (OCR) | top-1 | 58.33 (95.0%) | 42.86 (64.3%) | 49.92 (78.2%) | 45.82 (90.6%) | 13.90 (68.2%) | 47.12 (85.6%) | 42.99 (80.3%) |
| | top-2 | 59.44 (96.8%) | 47.62 (71.4%) | 56.51 (88.6%) | 47.08 (93.1%) | 15.87 (77.8%) | 51.08 (92.8%) | 46.27 (86.8%) |
| | top-3 | 61.76 (100.6%) | 44.44 (66.7%) | 55.67 (87.3%) | 49.58 (98.1%) | 14.72 (72.2%) | 49.28 (89.5%) | 45.91 (85.7%) |
| | Oracle | 61.40 (100%) | 66.67 (100%) | 63.79 (100%) | 50.56 (100%) | 20.39 (100%) | 55.04 (100%) | 52.97 (100%) |
| (b) VisRAG-Gen: *Single-image VLM (MiniCPM-V 2.0)* | | | | | | | | |
| Page Concatenation | top-1 | 59.07 (98.0%) | 34.92 (88.0%) | 39.42 (74.4%) | 29.53 (86.5%) | 17.84 (77.4%) | 36.15 (91.8%) | 36.16 (86.0%) |
| | top-2 | 57.35 (95.1%) | 19.05 (48.0%) | 32.32 (61.0%) | 22.14 (64.9%) | 15.41 (66.8%) | 33.45 (84.9%) | 29.95 (70.1%) |
| | top-3 | 59.19 (98.2%) | 22.22 (56.0%) | 24.87 (47.0%) | 20.33 (59.6%) | 16.92 (73.4%) | 30.22 (76.7%) | 28.96 (68.5%) |
| | Oracle | 60.29 (100%) | 39.68 (100%) | 52.96 (100%) | 34.12 (100%) | 23.06 (100%) | 39.39 (100%) | 41.58 (100%) |
| Weighted Selection | top-1 | 59.07 (98.0%) | 34.92 (88.0%) | 39.42 (74.4%) | 29.53 (86.5%) | 17.84 (77.4%) | 36.15 (87.4%) | 36.16 (85.3%) |
| | top-2 | 60.29 (100.0%) | 33.33 (84.0%) | 39.26 (74.1%) | 28.97 (84.9%) | 18.08 (78.4%) | 36.69 (88.7%) | 36.10 (85.0%) |
| | top-3 | 60.78 (100.8%) | 31.75 (80.0%) | 38.41 (72.5%) | 28.69 (84.1%) | 17.03 (73.9%) | 36.33 (87.8%) | 35.50 (83.2%) |
| | Oracle | 60.29 (100%) | 39.68 (100%) | 52.96 (100%) | 34.12 (100%) | 23.06 (100%) | 41.37 (100%) | 41.91 (100%) |
| (c) VisRAG-Gen: *Multi-image VLM* | | | | | | | | |
| MiniCPM-V 2.6 | top-1 | 66.30 (93.3%) | 47.62 (69.8%) | 60.24 (72.4%) | 56.41 (88.6%) | 40.79 (65.1%) | 48.56 (84.1%) | 53.32 (78.9%) |
| | top-2 | 66.79 (94.0%) | 52.38 (76.7%) | 67.17 (80.7%) | 53.90 (84.7%) | 38.35 (61.2%) | 50.90 (88.2%) | 54.92 (80.9%) |
| | top-3 | 67.77 (95.3%) | 53.97 (79.1%) | 70.90 (85.2%) | 54.46 (85.6%) | 38.93 (62.1%) | 50.72 (87.9%) | 56.12 (82.5%) |
| | Oracle | 71.08 (100%) | 68.25 (100%) | 83.25 (100%) | 63.65 (100%) | 62.69 (100%) | 57.73 (100%) | 67.78 (100%) |
| GPT-4o | top-1 | 64.71 (98.0%) | 52.38 (76.7%) | 58.88 (74.2%) | 63.09 (88.3%) | 20.74 (66.3%) | 54.86 (85.0%) | 52.44 (81.4%) |
| | top-2 | 63.36 (95.9%) | 49.21 (72.1%) | 64.13 (80.8%) | 66.85 (93.6%) | 20.16 (64.4%) | 58.45 (90.5%) | 53.69 (82.9%) |
| | top-3 | 62.01 (93.9%) | 53.97 (79.1%) | 67.17 (84.6%) | 66.43 (93.0%) | 19.35 (61.9%) | 60.97 (94.4%) | 54.98 (84.5%) |
| | Oracle | 66.05 (100%) | 68.25 (100%) | 79.36 (100%) | 71.45 (100%) | 31.29 (100%) | 64.57 (100%) | 63.49 (100%) |

Embed-v2 are trained on millions of query-doc pairs (Xiao et al., 2023; Lee et al., 2024), which are 10× more than our training data. Although bge-large outperforms BM25 on benchmarks like MTEB (Muennighoff et al., 2023), it fails on our datasets, indicating text-based embedding models trained on clean text struggle with texts parsed from real-world documents.

When trained with the same data setup, as demonstrated in Table 2(b)(c), VisRAG-Ret outperforms text models MiniCPM (OCR) & (Captioner) and the vision model SigLIP by a significant margin. The advantage is more pronounced in the out-of-domain setting, where VisRAG-Ret achieves 13% and 20% gains over MiniCPM (OCR) and SigLIP, respectively, compared to 4% and 9% in the in-domain setting. This indicates that VisRAG-Ret has better generalization capability compared to text- and vision-centric models. Notably, despite utilizing the same VLM MiniCPM-V 2.0 for parsing, MiniCPM (Captioner) performs worse than VisRAG-Ret, indicating that directly encoding with VLMs works better than using VLMs for parsing. This can be attributed to the inevitable information loss when multi-modality information is transcribed into text.

Further analysis reveals that MiniCPM (OCR) and SigLIP perform differently across datasets: SigLIP excels in ArxivQA and ChartQA, while MiniCPM (OCR) significantly outperforms SigLIP in DocVQA and InfographicsVQA. This may be due to the different focuses of the two models: MiniCPM focuses on text, while SigLIP focuses on visual signals. VisRAG-Ret, built on top of MiniCPM-V 2.0, with a SigLIP encoder and a MiniCPM language model, combines the merits of both and performs well across all datasets, capturing more holistic information from a document.

Compared to ColPali, a multi-vector document page embedding model, VisRAG-Ret not only maintains superior performance but also achieves much better memory efficiency. ColPali represents a page with 256KB of data distributed across 1030 128-dim vectors (Faysse et al., 2024), whereas VisRAG-Ret uses just 4.5KB in a single 2304-dimensional vector. This makes VisRAG-Ret more suitable for scaling to millions or billions of documents in real-world applications.

**Generation Performance.** In this experiment, we apply a series of text- and vision-based generators and methods on top of the same retriever VisRAG-Ret to study their effectiveness in generating the answer given the query and retrieved documents. Table 3 shows the performance of (a) text-based generation (TextRAG-Gen), (b) generation using the VLM MiniCPM-V 2.0 which only accepts a single image as input, and (c) generation using VLMs which accept multiple images as input.

When models are provided with only the ground-truth documents ("Oracle"), VisRAG-Gen models, which process the document image directly, significantly outperform TextRAG-Gen models, which

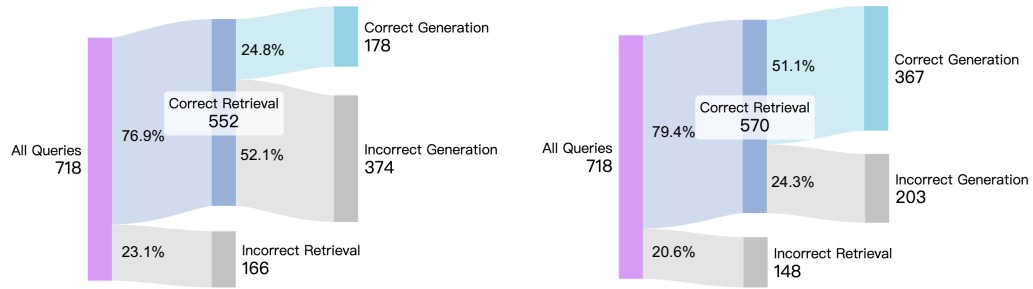

(a) TextRAG with MiniCPM (OCR) as the retriever and MiniCPM-V 2.6 (OCR) as the generator.

(b) VisRAG with VisRAG-Ret as the retriever and MiniCPM-V 2.6 as the generator.

Figure 3: Pipeline performance of (a) TextRAG and (b) VisRAG on InfographicsVQA. We visualize the portion of queries that have the positive document retrieved at the top-1 position ("Correct Retrieval"), and that are answered correctly given the top-1 retrieved document ("Correct Generation").

rely solely on extracted text. For instance, MiniCPM-V 2.0 achieves 30% higher performance than MiniCPM (OCR) when using ground-truth documents. This underscores the importance of visual clues in extracting answers from documents.

In practical scenarios where models receive the top-1 to 3 retrieved documents, which may include noise, VisRAG-Gen consistently outperforms TextRAG-Gen within the same model series. Specifically, for MiniCPM-V 2.0, capable of processing only a single image, the weighted selection approach demonstrates better performance than page concatenation when handling 2 or 3 retrieved documents. However, neither method shows a performance improvement as the number of retrieved documents increases, a trend commonly observed in TextRAG pipelines (Zhu et al., 2024). In contrast, MiniCPM-V 2.6 and GPT-4o, both capable of processing multiple images as input, exhibit a notable performance gain as the number of retrieved documents increases, suggesting that only VLMs pre-trained on multi-image data can effectively reason over multiple retrieved pages.

**End-to-end Performance.** In this experiment, we study the effectiveness of the VisRAG *pipeline*, by comparing it with the TextRAG pipeline. We construct TextRAG using MiniCPM (OCR) and MiniCPM-V 2.6 (OCR) for retrieval and generation, respectively, and VisRAG using VisRAG-Ret for retrieval and MiniCPM-V 2.6 for generation. The performance on InfographicsVQA is visually represented in Figure 3. Notably, VisRAG achieves a higher rate of accurately retrieving documents than TextRAG, and demonstrates a significantly improved rate of correct answer generation from accurately retrieved documents. The cumulative improvements in both retrieval and generation phases result in an overall accuracy increment from 25% to 51%. Across the six evaluation datasets, VisRAG shows a 40% relative accuracy increment on average, as illustrated in Figure 1. The case study of VisRAG and TextRAG is presented in Appendix F.

## 5.2 TRAINING DATA EFFICIENCY

In this experiment, we study the training data efficiency of VisRAG-Ret by evaluating the performance of VisRAG-Ret trained under different amounts of synthetic training data, i.e. in the out-of-domain setting. As shown in Figure 4, to achieve the same performance as bge-large (OCR), VisRAG-Ret requires training on only 20K examples, whereas MiniCPM (OCR) needs about 75K examples. In later training stages, VisRAG-Ret still maintains a 13% performance advantage over MiniCPM (OCR). Although NV-Embed-v2 (OCR) slightly outperforms VisRAG-Ret trained on our

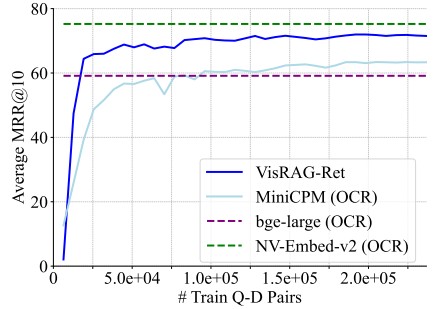

Figure 4: Average retrieval performance of VisRAG-Ret vs. MiniCPM (OCR) trained with different numbers of training examples.

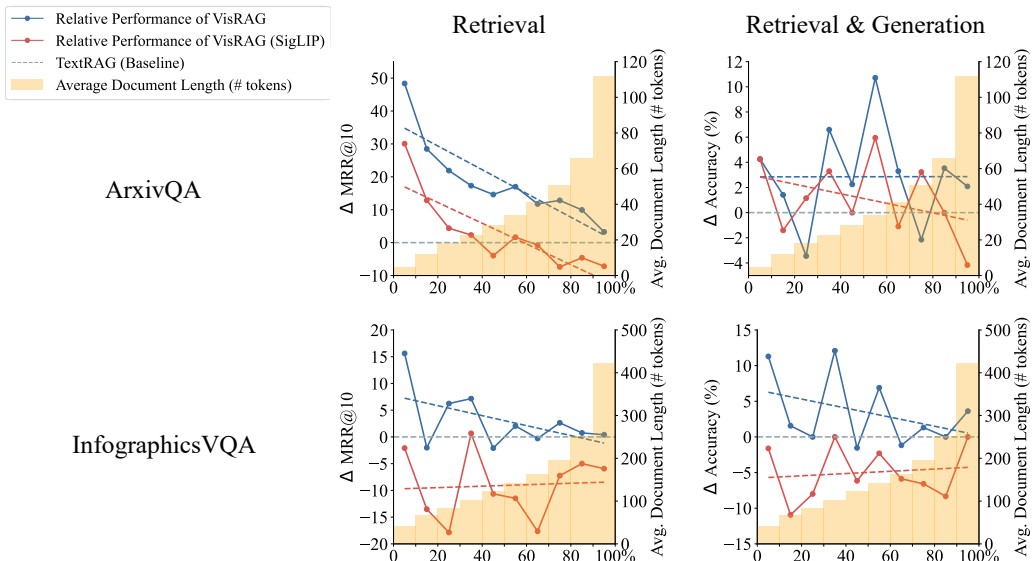

Figure 5: Relative retrieval and generation performance of VisRAG, VisRAG (SigLIP), and TextRAG on different subsets of queries. The X-axes represent the query subsets where the lengths of the positive documents fall within specific percentile ranges. For comparative analysis, we set TextRAG's performance to zero and show the performance differences of other models from TextRAG.

240K synthetic dataset, it is trained on millions of curated query-document pairs and has an 8B parameter scale. This suggests that capturing multi-modal information is more effective and efficient than merely increasing training data and model parameters but relying solely on the text modality.

## 5.3 PERFORMANCE ON DIFFERENT DATA SUBSETS

In this experiment, we assess the retrieval and generation performance of VisRAG and TextRAG defined in Figure 3, as well as VisRAG (SigLIP), which replaces the retriever in VisRAG with SigLIP. In Figure 5, we report their performance across different data subsets of ArxivQA and InfographicsVQA by categorizing queries based on the lengths of their positive documents, measured by the number of tokens of the extracted text. Documents with a higher volume of extracted text may prioritize textual information over visual content. For each group, we calculate and plot the average performance differences between VisRAG and TextRAG, as well as between VisRAG (SigLIP) and TextRAG, to compare how each model performs relative to TextRAG. We observe that, in general, the relative performance of VisRAG and VisRAG (SigLIP) improves as the length of the relevant document decreases. This suggests that models with vision encoders can better understand documents that emphasize visual information. However, VisRAG (SigLIP) consistently underperforms VisRAG across all data subsets and, in some cases, even performs worse than TextRAG. In contrast, VisRAG outperforms TextRAG on most subsets, indicating that the underlying language model in VisRAG is crucial for better understanding the semantics conveyed through visual cues.

## 6 CONCLUSION

In this paper, we propose VisRAG, a novel retrieval-augmented generation (RAG) paradigm that utilizes vision-language models (VLMs) to facilitate retrieval and generation within an RAG pipeline, thereby eliminating the parsing stage required in traditional text-based RAG. Our empirical results demonstrate that VisRAG consistently outperforms text-based RAG on retrieval and generation while maintaining a simpler pipeline. We hope that VisRAG will inspire future RAG development to incorporate VLMs for handling multi-modal documents.

ACKNOWLEDGMENTS

This work is supported by the Institute Guo Qiang at Tsinghua University. It is also partially supported by the Natural Science Foundation of China under Grant No. 62206042.

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

# A  DATA CONSTRUCTION DETAILS

## A.1  SYNTHETIC DATA

Table 4: Statistics of crawled documents. We prompt GPT-4o to generate queries on these documents.

| Name | Source | Description | # Pages |
|------|--------|-------------|---------|
| Textbooks | https://openstax.org/ | College-level textbooks including various subjects | 10,000 |
| ICML Papers | ICML 2023 | ICML papers on various topics | 5,000 |
| NeurIPS Papers | NeurIPS 2023 | NeurIPS papers on various topics | 5,000 |
| Manuallib | https://www.manualslib.com/ | Manuals of various kinds of products | 20,000 |

To augment the training dataset of VisRAG, we gather additional documents from the web and utilize GPT-4o to generate queries based on these documents. The sources of the collected documents are listed in Table 4. The prompt employed is shown in Figure 6.

```
Hello, I have a super rich document library. Assume you are a curious but very ignorant
    human. You often ask me questions (queries) to seek a precise document as a
    reference for your question or request.
- Now, you have received another task:
    - Here is a document image. This is a reference (target) that I provided from the
    rich document library based on your query. Your task now is to imagine various
    different angles of questions that I might ask.
### Your goal is to accurately find this document target as a potential reference
    document candidate through queries in a very rich document library.
### The questions I ask might need references from the text, images, charts, or
    implicit meanings in the document.
### Maximum number of query-answer pairs is 6.

Below is your output format:
```json
{
    "result":[
        {
            "answer": "",
            "query" : ""
        },
        {
            "answer": "",
            "query" : ""
        },
    ...
        {
            "answer": "",
            "query" : ""
        }
    ]
}
```
{{ document }}
```

Figure 6: Prompt for GPT-4o to generate queries, where {{ document }} is the document page.

## A.2  QUERY FILTERING

As mentioned in Sec. 3.3, a significant portion of queries in VQA datasets are context-dependent and thus unsuitable for retrieval. To filter out such queries, we prompt GPT-4o (OpenAI, 2024) using the instruction shown in Figure 7, which includes human-annotated samples from DocVQA. Although this filtering step reduces context-dependent queries, a small number may still remain. However, their presence is minimal and does not significantly impact the overall quality of our dataset.

```
I have some QA data here, and you can observe that the questions can be divided into
    two categories:

The category #A: When you see this question alone without a given document, you are
    sure to find a unique document in a corpus to provide a unique answer.

The category #B: When you see this question alone without a given document, you will
    find hard to locate a document to give a deterministic answer for this question,
    because you will find multiple candidate documents in a corpus, which may lead to
    different answers for this question.

Here are some examples:
The number mentioned on the right of the leftside margin? #B
What is the date mentioned in the second table? #B
What is the full form of PUF? #A
What is the number at the bottom of the page, in bold? #B
Who presented the results on cabin air quality study in commercial aircraft? #A
What is the name of the corporation? #B
To whom this is addressed? #B
How many one-on-one interviews were completed during April 10th through the April 12th?
    #A
What is the subject of the document/letter? #B
Who sent the letter? #B
Heading of the document? #B
What is the slope mentioned in the first table? #B
what is the date in the letter? #B
What is the date mentioned in the letter? #B
Which part of Virginia is this letter sent from? #B
who were bothered by cigarette odors? #A
which cigarette would be better if offered on a thicker cigarette? #A
Cigarettes will be produced and submitted to O/C Panel for what purpose? #A
What is the heading of first table? #B
What is RIP-6 value for KOOL KS? #A
Which hetero-atoms does polar compounds contain? #A
One variable that has implicitly not been controlled? #B
Which corporation's letterhead is this? #B
what is the contact person name mentioned in letter? #B
what is the date mentioned in this letter? #B
Another model of the 83mm with zero ventilation will be made at Semiworks within how
    many weeks? #A
Hand sheets were made utilizing a 30% level of which component? #A
What is the source? #B
What is the heading of the document? #B
What is the subject? #B
What is the S.D. mentioned in the DOSE-ug 0.0000 in the third table? #B
Which base paper will be coated in-house with various levels of mono potassium
    phosphate and malonic acid in order to optimize the system? #A
Which test is used to evaluate ART menthol levels that has been shipped? #A
How much percent had not noticed any difference in the odor of VSSS? #A
What is the cigarette code of RIP-6(W/O Filter) 21/4SE? #A
What is the meeting date? #B
How many points are there in modifications to readout instrumentation? #A
what is the subject of this letter? #B
what is the index for Retention of Franchise? #B
What is the heading of second table? #B
What is the full form of POVC? #A
what mm Marlboro Menthol were subjectively smoked by the Richmond Panel? #A
What sort of communication/letter is this? #B
How many one-on-one interviews were completed during April 10th through the April 12th?
    #A
During the process of prototype production and ringtipping, some cigarettes were
    observed to have burn holed in which paper? #A
How many distinct mechanisms appear to play a role in the breakup of a smoke column
    into a multi-dimensional flowfield? #A
Where was the conference held? #B
Who is in cc in this letter? #B
Under BOLD, primary production of Blend #24- will be completed by which date? #A

Query: {{ query }}
Determine if the query belongs to Category #A or Category #B.
Output only A or B.
```

Figure 7: Prompt for GPT-4o to classify queries, where {{ query }} is the query to be classified. Label B denotes context-dependent queries.

## B    DOCUMENT PARSING

In this paper, we experiment with two categories of document parsing strategies: pipeline-based parsing and model-based parsing.

### B.1    PIPELINE-BASED PARSING

We consider the following document parsing pipelines:

**Pytesseract.**   Pytesseract is a Python wrapper for Google's Tesseract OCR engine, offering a straightforward interface for text extraction from images. Unlike more complex methods, Pytesseract requires minimal pre-processing. By invoking the `image_to_string` function, OCR is performed in a single step, directly returning the extracted text. Tesseract internally handles bounding boxes, confidence scores, and orientation correction.

**PPOCR-based Methods.**   PaddlePaddle OCR (PPOCR) (Du et al., 2020) is widely used for document text extraction, covering text detection, classification, and recognition. First, a text detection model identifies text regions and generates bounding boxes. These regions are then processed by a classification model to correct orientation issues like rotation or flipping. Next, a recognition model extracts the textual content from the corrected bounding boxes, returning recognized text with confidence scores. Only results with confidence scores above 0.6 are retained, and the bounding box coordinates, along with the recognized text, are stored for further processing. We apply the following strategies to obtain the final parsing result:

- Adjacent Merging: To enhance text coherence, this policy combines adjacent text boxes based on vertical proximity (within 15 pixels) and horizontal alignment (within 100 pixels), reducing text fragmentation. This iterative merging process consolidates eligible text boxes into unified bounding boxes with concatenated text. Finally, the text from the remaining bounding boxes is combined with line breaks to produce the final result.

- Layout Preserving: This policy maintains the original document structure by ordering text boxes based on their spatial positions. Spaces and line breaks are dynamically inserted to reflect horizontal and vertical gaps between text regions. This approach ensures that the extracted text mirrors the original document layout, preserving its formatting in the final result.

We run the aforementioned pipelines on our dataset to obtain text-based training and evaluation data, and fine-tune a MiniCPM retriever to assess performance. The results are presented in Table 5. Methods based on PPOCR demonstrate significantly better performance compared to pytesseract, with adjacent merging and layout preserving yielding similar results. Consequently, we opt to use the adjacent merging policy for our "(OCR)" runs.

Table 5: Overall retrieval performance of different document parsing pipelines.

|  | ArxivQA | ChartQA | DocVQA | InfoVQA | PlotQA | SlideVQA | Average |
|---|---|---|---|---|---|---|---|
| (c) In-domain: *Models Fine-tuned on Synthetic and In-domain data* | | | | | | | |
| MiniCPM (Pytesseract) | 41.53 | 72.40 | 70.67 | 76.45 | 55.96 | 79.79 | 66.13 |
| MiniCPM (Adjacent Merging) | **58.43** | **77.74** | **72.54** | **83.45** | **64.78** | **91.74** | **74.78** |
| MiniCPM (Layout Preserving) | 55.81 | 75.40 | 71.70 | 83.12 | 63.65 | 91.64 | 73.55 |

### B.2    MODEL-BASED PARSING

In addition to pipeline-based methods, we also employ a model-based parsing approach using MiniCPM-V 2.0 to directly transcribe document images into text. This method is referred to as "(Captioner)".

To train this model, we collect data from two sources: a) ALLaVA (Chen et al., 2024a) (image, caption) pairs, and b) VQA documents with descriptions generated by GPT-4V. We use the prompt in Figure 8 to instruct GPT-4V to generate detailed descriptions of documents from DocVQA, ChartQA, SlideVQA, InfographicsVQA, TextVQA (Singh et al., 2019), and ArxivQA.

```
Based on the layout information, output the text in the image. Try not to modify the
    text, but you need to indicate the structure such as title, body text, subtitle,
    table, etc.
Note:
If there are charts or graphs, they should be described in detail.
If you feel that there are more than 4000 words or most of the text in the image is
    unclear or most of the text contents in the image are not written in English, then
     directly return <none>.
{{ document }}
```

Figure 8: Prompt for GPT-4V to generate page description, where {{ document }} is the document page.

We train MiniCPM-V 2.0 with a batch size of 2048 and a learning rate of 5e-6 for 1 epoch.

## C  MODELS USED IN THIS PAPER

**MiniCPM**   (Hu et al., 2024c) is a large language model (LLM) with 2.4 billion non-embedding parameters, demonstrating capabilities comparable to much larger models, such as Llama2-7B (Touvron et al., 2023) and Gemma-7B (Team et al., 2024). In this paper, we employ MiniCPM to construct the baseline text-based retriever (Table 2) and generator (Table 3).

**SigLIP**   (Zhai et al., 2023) is a CLIP-style multi-modal model designed to align text and vision representations. We utilize SigLIP-400m, released by Hugging Face[2], which incorporates Flash Attention 2, increases maximum resolution to 980x980, and adopts the NaViT strategy to allow (a) variable resolution images and (b) aspect ratio preserved images. In this paper, SigLIP is used to develop the baseline vision-based retriever (Table 2).

**MiniCPM-V 2.0**   (OpenBMB, 2024a; Yao et al., 2024) is a vision-language model (VLM) with 2.8 billion non-embedding parameters, built upon SigLIP-400m and MiniCPM. It can process single images up to 1.8 million pixels (e.g., 1344x1344) at any aspect ratio. We use MiniCPM-V 2.0 to build VisRAG-Ret (Table 2) and VisRAG-Gen (Table 3(b)), as well as the document parsing model.

**MiniCPM-V 2.6**   (OpenBMB, 2024b; Yao et al., 2024) is an upgrade of MiniCPM-V 2.0 and MiniCPM-Llama3-V 2.5 (Yao et al., 2024). It is built upon SigLIP-400M and Qwen2-7B (Yang et al., 2024) with a total of 8.5B parameters, exihibiting a significant performance improvement over MiniCPM-Llama3-V 2.5 (Yao et al., 2024). Different from previous models, MiniCPM-V 2.6 can accept multiple images as the input and perform multi-modal in-context learning. It also demonstrates stronger OCR capabilities. We use MiniCPM-V 2.6 to build VisRAG-Gen (Table 3) and a text-based generation baseline MiniCPM-V 2.6 (OCR) (Figure 3, Figure 5).

Note that, MiniCPM-Llama3-V 2.5 (Yao et al., 2024) is not used in this paper.

**GPT-4o**   (OpenAI, 2024) is OpenAI's latest multi-modal model, capable of processing any combination of text, audio, image, and video inputs and generating outputs in text, audio, and image formats. We use GPT-4o to construct VisRAG-Gen (Table 3) and to synthesize training data.

## D  RETRIEVAL PERFORMANCE IN RECALL@10

Table 6 presents the retrieval performance in Recall@10.

## E  PROMPTS FOR GENERATION

We present the prompts of VisRAG-Gen and TextRAG-Gen in Table 7.

---

[2]https://huggingface.co/HuggingFaceM4/siglip-so400m-14-980-flash-attn2-navit

Table 6: Overall retrieval performance in Recall@10.

| Model | # Para. | ArxivQA | ChartQA | DocVQA | InfoVQA | PlotQA | SlideVQA | Average |
|---|---|---|---|---|---|---|---|---|
| (a) Off-the-shelf Models | | | | | | | | |
| BM25 (OCR) | n.a. | 54.29 | 79.37 | 86.80 | 82.59 | 76.01 | 91.64 | 78.45 |
| bge-large (2023) (OCR) | 335M | 48.65 | 76.19 | 68.19 | 88.16 | 73.12 | 90.11 | 74.07 |
| NV-Embed-v2 (2024) (OCR) | 7.85B | **70.10** | **88.89** | **89.85** | **95.13** | **80.88** | **97.84** | **87.11** |
| SigLIP (2023) | 883M | 44.98 | 77.78 | 68.02 | 84.68 | 58.29 | 89.03 | 70.46 |
| (b) Out-of-domain: *Models Fine-tuned on Synthetic Data* | | | | | | | | |
| MiniCPM (OCR) | 2.72B | 59.07 | 79.37 | 84.26 | 91.64 | 60.25 | 94.78 | 78.23 |
| MiniCPM (Captioner) | 2.72B | 55.64 | 82.54 | 79.19 | 92.06 | 57.71 | 90.11 | 76.21 |
| SigLIP (2023) | 883M | 60.17 | 82.54 | 75.47 | 84.82 | 59.33 | 92.81 | 75.85 |
| VisRAG-Ret | 3.43B | **81.00** | **84.13** | **87.65** | **97.08** | **71.84** | **95.59** | **86.22** |
| (c) In-domain: *Models Fine-tuned on Synthetic and In-domain data* | | | | | | | | |
| MiniCPM (OCR) | 2.72B | 69.36 | 88.89 | 87.14 | 94.15 | **90.61** | 96.85 | 87.83 |
| MiniCPM (Captioner) | 2.72B | 69.00 | 85.71 | 84.26 | 94.29 | 84.24 | 93.08 | 85.10 |
| SigLIP (2023) | 883M | 73.90 | **92.06** | 83.08 | 93.04 | 89.57 | 94.15 | 87.63 |
| ColPali (2024) | 2.92B | 82.72 | 88.89 | **94.75** | 94.43 | 80.30 | 97.21 | 89.72 |
| VisRAG-Ret | 3.43B | **87.25** | 90.48 | 91.20 | **97.08** | 89.80 | 97.39 | **92.20** |

Table 7: Prompt templates for generation. "Others" refers to all VQA datasets except ArxivQA.

| | **TextRAG** | **VisRAG** |
|---|---|---|
| **ArxivQA** | Hint: {{ parsed document(s) }}
Question: {{ query }}
Options:
A. {{ Option 1 }}
B. {{ Option 2 }}
C. {{ Option 3 }}
D. {{ Option 4 }}
Answer directly with the letter of the correct option as the first character. | {{ document(s) }}
Question: {query }}
Options:
A. {{ Option 1 }}
B. {{ Option 2 }}
C. {{ Option 3 }}
D. {{ Option 4 }}
Answer directly with the letter of the correct option as the first character. |
| **Others** | Image:{{ parsed document(s) }}
Answer the question using a single word or phrase.
Question:{{ query }}
Answer: | {{ document(s) }}
Answer the question using a single word or phrase.
Question:{{ query }}
Answer: |

# F  CASE STUDY

We show two cases in Table 8 and Table 9. In both instances, we compare VisRAG with TextRAG, maintaining the same setup as described in the "End-to-end Performance" paragraph in Sec. 5.1.

In the first case from DocVQA, the user queries about "Club Jetty," however, the term "Club Jetty" in the relevant document is not successfully extracted due to its decorative font. This leads to TextRAG failing to retrieve the document, while VisRAG successfully retrieves it.

In the second case from InfographicsVQA, although both TextRAG and VisRAG successfully retrieve the document, TextRAG generates an incorrect response due to the loss of layout information, making it unclear which number (53% or 49%) pertains to Europe. VisRAG effectively utilizes the layout information and generates the correct answer.

Table 8: Case study from DocVQA. In this case, VisRAG successfully retrieves the ground-truth document, while TextRAG fails, leading to VisRAG's correct generation and TextRAG's incorrect generation.

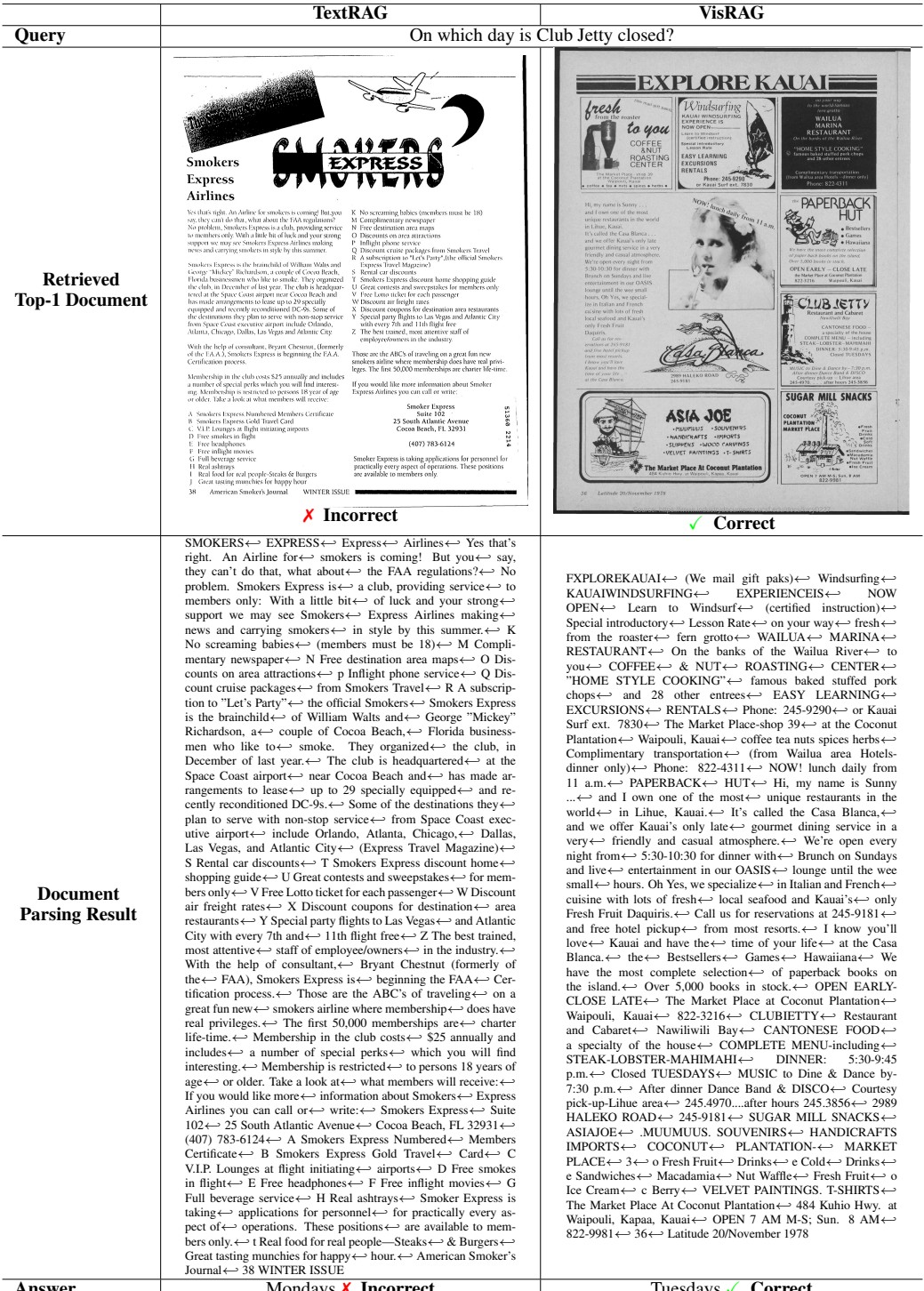

| | TextRAG | VisRAG |
|---|---|---|
| **Query** | On which day is Club Jetty closed? | |
| **Retrieved Top-1 Document** | ✗ **Incorrect** | ✓ **Correct** |
| **Document Parsing Result** | SMOKERS← EXPRESS← Express← Airlines← Yes that's right. An Airline for← smokers is coming! But you← say, they can't do that, what about← the FAA regulations?← No problem. Smokers Express is← a club, providing service← to members only: With a little bit← of luck and your strong← support we may see Smokers← Express Airlines making← news and carrying smokers← in style by this summer.← K No screaming babies← (members must be 18)← M Complimentary newspaper← N Free destination area maps← O Discounts on area attractions← p Inflight phone service← Q Discount cruise packages← from Smokers Travel← R A subscription to "Let's Party"← the official Smokers← Smokers Express is the brainchild← of William Walts and← George "Mickey" Richardson, a← couple of Cocoa Beach,← Florida businessmen who like to← smoke. They organized← the club, in December of last year.← The club is headquartered← at the Space Coast airport← near Cocoa Beach and← has made arrangements to lease← up to 29 specially equipped← and recently reconditioned DC-9s.← Some of the destinations they← plan to serve with non-stop service← from Space Coast executive airport← include Orlando, Atlanta, Chicago,← Dallas, Las Vegas, and Atlantic City← (Express Travel Magazine)← S Rental car discounts← T Smokers Express discount home← shopping guide← U Great contests and sweepstakes← for members only← V Free Lotto ticket for each passenger← W Discount air freight rates← X Discount coupons for destination← area restaurants← Y Special party flights to Las Vegas← and Atlantic City with every 7th and← 11th flight free← Z The best trained, most attentive← staff of employee/owners← in the industry.← With the help of consultant,← Bryant Chestnut (formerly of the← FAA), Smokers Express is← beginning the FAA← Certification process.← Those are the ABC's of traveling← on a great fun new← smokers airline where membership← does have real privileges.← The first 50,000 memberships are← charter life-time.← Membership in the club costs← $25 annually and includes← a number of special perks← which you will find interesting.← Membership is restricted← to persons 18 years of age← or older. Take a look at← what members will receive:← If you would like more← information about Smokers← Express Airlines you can call or← write:← Smokers Express← Suite 102← 25 South Atlantic Avenue← Cocoa Beach, FL 32931← (407) 783-6124← A Smokers Express Numbered← Members Certificate← B Smokers Express Gold Travel← Card← C V.I.P. Lounges at flight initiating← airports← D Free smokes in flight← E Free headphones← F Free inflight movies← G Full beverage service← H Real ashtrays← Smoker Express is taking← applications for personnel← for practically every aspect of← operations. These positions← are available to members only.← t Real food for real people—Steaks← & Burgers← Great tasting munchies for happy← hour.← American Smoker's Journal← 38 WINTER ISSUE | FXPLOREKAUAI← (We mail gift paks)← Windsurfing← KAUAIWINDSURFING← EXPERIENCEIS← NOW OPEN← Learn to Windsurf← (certified instruction)← Special introductory← Lesson Rate← on your way← fresh← from the roaster← fern grotto← WAILUA← MARINA← RESTAURANT← On the banks of the Wailua River← to you← COFFEE← & NUT← ROASTING← CENTER← "HOME STYLE COOKING"← famous baked stuffed pork chops← and 28 other entrees← EASY LEARNING← EXCURSIONS← RENTALS← Phone: 245-9290← or Kauai Surf ext. 7830← The Market Place-shop 39← at the Coconut Plantation← Waipouli, Kauai← coffee tea nuts spices herbs← Complimentary transportation← (from Wailua area Hotels-dinner only)← Phone: 822-4311← NOW! lunch daily from 11 a.m.← PAPERBACK← HUT← Hi, my name is Sunny ...← and I own one of the most← unique restaurants in the world← in Lihue, Kauai.← It's called the Casa Blanca,← and we offer Kauai's only late← gourmet dining service in a very← friendly and casual atmosphere.← We're open every night from← 5:30-10:30 for dinner with← Brunch on Sundays and live← entertainment in our OASIS← lounge until the wee small← hours. Oh Yes, we specialize← in Italian and French← cuisine with lots of fresh← local seafood and Kauai's← only Fresh Fruit Daquiris.← Call us for reservations at 245-9181← and free hotel pickup← from most resorts.← I know you'll love← Kauai and have the← time of your life← at the Casa Blanca.← the← Bestsellers← Games← Hawaiiana← We have the most complete selection← of paperback books on the island.← Over 5,000 books in stock.← OPEN EARLY-CLOSE LATE← The Market Place at Coconut Plantation← Waipouli, Kauai← 822-3216← CLUBIETTY← Restaurant and Cabaret← Nawiliwili Bay← CANTONESE FOOD← a specialty of the house← COMPLETE MENU-including← STEAK-LOBSTER-MAHIMAHI← DINNER: 5:30-9:45 p.m.← Closed TUESDAYS← MUSIC to Dine & Dance by-7:30 p.m.← After dinner Dance Band & DISCO← Courtesy pick-up-Lihue area← 245.4970....after hours 245.3856← 2989 HALEKO ROAD← 245-9181← SUGAR MILL SNACKS← ASIAJOE← .MUUMUUS. SOUVENIRS← HANDICRAFTS IMPORTS← COCONUT← PLANTATION-← MARKET PLACE← 3←o Fresh Fruit← Drinks←e Cold← Drinks← e Sandwiches← Macadamia← Nut Waffle← Fresh Fruit← o Ice Cream← c Berry← VELVET PAINTINGS. T-SHIRTS← The Market Place At Coconut Plantation← 484 Kuhio Hwy. at Waipouli, Kapaa, Kauai← OPEN 7 AM M-S; Sun. 8 AM← 822-9981← 36← Latitude 20/November 1978 |
| **Answer** | Mondays ✗ **Incorrect** | Tuesdays ✓ **Correct** |

Table 9: Case study from InfographicsVQA. In this case, both VisRAG and TextRAG successfully retrieve the correct document; however, only VisRAG effectively leverages the layout information, enabling accurate generation. In contrast, TextRAG suffers from information loss of the layout, resulting in incorrect responses.

|  | **TextRAG** | **VisRAG** |
|---|---|---|
| **Query** | What percent of account holders in Europe are using LinkedIn for finding job? | |
| **Retrieved Top-1 Document** |  ✓ **Both Correct** | |
| **Document Parsing Result** | Social media← job seeking trends← Michael Page's annual global survey of financial services and banking← employees was conducted in April 2014,more than 3,300 people participated← LinkedIn← Linkedin's popularity continues to grow, though many job seekers don't think of it as part of← their strategy.So hirers need to look to other sourcing channels too← What proportion of account holders← use Linkedin for job seeking?← 93← %← 30%← of respondents have← anaccount-up← 10% from last year← more women← than men say← they don't have← an account← 53%← In Europe← 49%← In North America← 40%← In the UK← Facebook← Despite last year's hype around Graph Search,Facebook hasn't made any progress with monetising← its recruitment potential -jobseekers remain very negative about Facebook playing any part← 13%← said they'd be happy← to see adverts← 92%← said they would not be← happy to be contacted by← a recruiter on Facebook← 1%← Don't bank on social media – Michael Page brings you a broader range of talent, and jobs← www.michaelpage.com.au/salarycentre← of respondents← (who are job seekers) said they← would use it to look for jobs← MichaelPage← Financial Services← Specialists in financial services recruitment← www.michaelpage.com.au← | |
| **Answer** | 49% ✗ **Incorrect** | 53% ✓ **Correct** |

## G  ADDITIONAL RETRIEVAL AND GENERATION RESULTS

Table 10: Additional retrieval performance in MRR@10.

| Model | ArxivQA | ChartQA | DocVQA | InfoVQA | PlotQA | SlideVQA | Average |
|---|---|---|---|---|---|---|---|
| (b) Out-of-domain: *Models Fine-tuned on Synthetic Data* | | | | | | | |
| MiniCPM (OCR) | 47.96 | 61.64 | 67.04 | 79.36 | 36.04 | 87.93 | 63.33 |
| SigLIP (2023) | 46.81 | 68.40 | 57.61 | 67.12 | 31.92 | 85.14 | 59.50 |
| MiniCPM (OCR) + SigLIP (RRF) | 54.07 | 72.33 | 65.46 | 75.32 | 38.98 | 88.06 | 65.70 |
| (c) In-domain: *Models Fine-tuned on Synthetic and In-domain data* | | | | | | | |
| MiniCPM (OCR) | 58.43 | 77.74 | 72.54 | 83.45 | 64.78 | 91.74 | 74.78 |
| SigLIP (2023) | 59.16 | 81.34 | 64.60 | 74.59 | 61.32 | 89.08 | 71.68 |
| MiniCPM (OCR) + SigLIP (RRF) | 64.19 | 85.39 | 71.75 | 80.88 | 66.09 | 92.94 | 76.87 |

Table 11: Additional generation performance in accuracy (%). All models and methods utilize the same retriever, VisRAG-Ret. Performance relative to Oracle is colored in blue.

| Model / Method | Input | ArxivQA | ChartQA | DocVQA | InfoVQA | PlotQA | SlideVQA | Average |
|---|---|---|---|---|---|---|---|---|
| (b) VisRAG-Gen: *Single-image VLM (MiniCPM-V 2.0)* | | | | | | | | |
| Page Concatenation | top-6 | 59.19 (98.2%) | 22.22 (56.0%) | 14.72 (27.8%) | 15.60 (45.7%) | 16.80 (72.9%) | 23.92 (60.7%) | 25.41 (60.2%) |
| | top-10 | 56.74 (94.1%) | 20.63 (52.0%) | 10.32 (19.5%) | 13.93 (40.8%) | 17.15 (74.4%) | 22.84 (58.0%) | 23.60 (56.5%) |
| | Oracle | 60.29 (100%) | 39.68 (100%) | 52.96 (100%) | 34.12 (100%) | 23.06 (100%) | 39.39 (100%) | 41.58 (100%) |
| (c) VisRAG-Gen: *Multi-image VLM* | | | | | | | | |
| MiniCPM-V 2.6 | top-6 | 67.89 (95.5%) | 57.14 (83.7%) | 70.05 (84.1%) | 51.25 (80.5%) | 35.81 (57.1%) | 51.80 (89.7%) | 55.66 (81.8%) |
| | top-10 | 64.95 (91.4%) | 57.14 (83.7%) | 54.48 (65.4%) | 36.49 (57.3%) | 30.94 (49.4%) | 51.80 (89.7%) | 49.30 (72.8%) |
| | Oracle | 71.08 (100%) | 68.25 (100%) | 83.25 (100%) | 63.65 (100%) | 62.69 (100%) | 57.73 (100%) | 67.78 (100%) |
| Qwen2-VL | top-1 | 66.30 (94.7%) | 53.97 (73.9%) | 65.82 (75.5%) | 55.71 (86.4%) | 51.33 (64.0%) | 55.58 (85.1%) | 58.12 (80.0%) |
| | top-2 | 65.44 (93.5%) | 52.38 (71.7%) | 70.90 (81.4%) | 55.15 (85.5%) | 47.05 (58.7%) | 58.99 (90.4%) | 58.32 (80.2%) |
| | top-3 | 67.03 (95.8%) | 57.14 (78.3%) | 73.60 (84.5%) | 52.79 (81.9%) | 44.96 (56.1%) | 58.63 (89.8%) | 59.03 (81.0%) |
| | Oracle | 69.98 (100%) | 73.02 (100%) | 87.14 (100%) | 64.48 (100%) | 80.19 (100%) | 65.29 (100%) | 73.35 (100%) |

In this section, we present supplementary evaluation results for both retrieval and generation on our dataset.

Table 10 shows additional retrieval results obtained by applying reciprocal rank fusion (RRF) (Cormack et al., 2009) to combine the outputs of MiniCPM (OCR) and SigLIP. It is a straightforward method to integrate textual information extracted from the page with its visual clues. The results indicate that fusing text and image modalities provides a meaningful performance boost over individual modality baselines. However, this approach still falls short of the performance achieved by our VisRAG-Ret model (71.49 for out-of-domain, 77.91 for in-domain). This underscores the superior capability of VisRAG-Ret in understanding both modalities within a unified architecture.

Table 11 provides additional generation results using top-6 and top-10 retrieved documents from VisRAG-Ret. For these experiments, we evaluate the performance of MiniCPM-V 2.0 using the page concatenation method and MiniCPM-V 2.6 with direct feeding. We also report the performance of another SOTA VLM, Qwen2-VL-7B-Instruct (Wang et al., 2024). The results indicate significant performance degradation when handling a larger number of retrieved pages, for both page concatenation (MiniCPM-V 2.0) and multi-page input (MiniCPM-V 2.6). MiniCPM-V 2.6 exhibits greater robustness to increasing context compared to MiniCPM-V 2.0. Open-source VLMs still face challenges in reasoning over multiple pages and extracting relevant information from noisy retrieved data. Results for Qwen2-VL demonstrate stronger document understanding capabilities, outperforming MiniCPM-V 2.6 in these tasks.

## H  RETRIEVAL EFFICIENCY

In this experiment, we evaluate the retrieval efficiency of VisRAG-Ret and MiniCPM (OCR) by measuring two key components: offline document parsing and encoding latency, and online query encoding and search latency. Query and document encoding are conducted on an NVIDIA A100 40G GPU with a batch size of 1, while document parsing is performed on a single core of an Intel Xeon Platinum 8350C CPU. The reported latencies are averaged over the queries and documents from the PlotQA dataset. The results are summarized in Table 12.

As shown in the table, although VisRAG-Ret, a VLM-based model, requires more time for document encoding compared to MiniCPM (OCR), it bypasses the time-consuming parsing stage required by

Table 12: Retrieval efficiency (ms). We report offline latencies per document, including document parsing and encoding latencies, as well as online latencies per query, including query encoding and search latencies.

| | Offline Latency per Document | | | Online Latency per Query | | |
|---|---|---|---|---|---|---|
| | Parsing | Encoding | Total | Encoding | Search | Total |
| MiniCPM (OCR) | 284 | 28 | 312 | 28 | 26 | 54 |
| VisRAG-Ret | – | 121 | 121 | 28 | 26 | 54 |

MiniCPM (OCR). This leads to a 58% reduction in total document processing time for VisRAG-Ret. For online query processing, the latencies of VisRAG-Ret and MiniCPM (OCR) are nearly identical, as the queries consist solely of textual inputs.

# I  RETRIEVAL PERFORMANCE ON TEXT RETRIEVAL BENCHMARKS

Table 13: Retrieval performance on subsets of the text retrieval benchmark BEIR (Thakur et al., 2021) in NDCG@10. VisRAG-Ret performs retrieval on rendered document screenshots.

| Model | SciFact | NFCorpus | Scidocs |
|---|---|---|---|
| MiniCPM (OCR) | 61.04 | 14.12 | 13.01 |
| VisRAG-Ret | 62.47 | 27.02 | 16.25 |

To evaluate how VisRAG-Ret performs in retrieval scenarios involving only textual data, we conduct an experiment using the BEIR (Thakur et al., 2021) text retrieval benchmark. To evaluate VisRAG-Ret, we convert the document texts into rendered screenshots and apply VisRAG-Ret to this modified dataset. We use the Pillow[3] library to convert text documents into screenshots, setting a width of 800px, a font size of 24px, and the DejaVuSans font. The height of each screenshot varies depending on the document length, with a margin of 20px and a line spacing of 4px. For comparison, we include MiniCPM (OCR) in the evaluation, utilizing raw textual data directly available in BEIR. Note that the term "OCR" in MiniCPM (OCR) is used solely for naming consistency.

As shown in Table 13, VisRAG-Ret, relying only on the rendered screenshots, significantly outperforms MiniCPM (OCR) which uses textual information. This result highlights that VisRAG-Ret's pooling-based representation effectively captures textual details and is well-suited for text-heavy document retrieval.

---

[3]https://python-pillow.org/

