# OpenReview forum: "VisRAG: Vision-based Retrieval-augmented Generation on Multi-modality Documents"
_ICLR.cc/2025/Conference — ICLR 2025 Poster_

### Official Review · Reviewer_kCGC · 2024-11-02

**Soundness:** 2
**Presentation:** 3
**Contribution:** 2
**Rating:** 6
**Confidence:** 5

**Summary:**

This paper proposes VisRAG, a model that leverages vision-language models (VLMs) for retrieval-augmented generation in multi-modal documents. The framework comprises two components: VisRAG-Ret for retrieval and VisRAG-Gen for generation, utilizing a data construction method that combines visual question answering (VQA) datasets with synthetic data. Experiments demonstrate that VisRAG outperforms traditional RAG in both retrieval and generation tasks, exhibiting improved training data efficiency and generalization capabilities. It achieves a significant relative increase in end-to-end accuracy, highlighting its potential to replace text-based RAG for multi-modal documents.

**Strengths:**

1.  The paper provides a clear and detailed description of the VisRAG framework., which makes it easy to understand.
2. The experimental results on document-related VQA tasks are promising, suggesting that the proposed method could be a practical solution.

**Weaknesses:**

1. The rationale behind the proposed VisRAG-Ret is unclear. If no OCR information is included, how can the target image be retrieved, especially for answers that require precise text content? From my perspective, a pooling-based representation feature does not offer sufficient information.

2. The experimental setup is not sufficiently convincing, indicating that the VQA task alone cannot fully demonstrate the method's effectiveness in the retrieval task. In other words, employing such a complex pipeline to address the VQA task is akin to using a sledgehammer to crack a nut.

3. The methods discussed are insufficient. To the best of my knowledge, there are several document-specific MLLMs, such as mlug-docowl, ureader, and text monkey, among others. While these works may not be directly related to RAG, they should at least be mentioned and analysized.

4. In my view, the comparison is unfair, as both Minicpm and Siglip can not only perform VQA tasks but also handle other tasks related to document image understanding and content perception.

**Questions:**

See "weaknesses"

---

> ### Author Response · Authors · 2024-11-19
>
> Thank you for your detailed review. First we would like to clarify an important point about our work that our focus is not on addressing the visual question answering (VQA) task. Instead, our work targets a retrieval-augmented generation (RAG) task on multi-modal documents, as described in Section 3 of our paper. In this pipeline, the system receives a user query, retrieves relevant documents from the corpus, and then generates an answer based on the retrieved information.
>
> Please find our point-by-point response to your comments/questions below.
>
> > Weakness 1. The rationale behind the proposed VisRAG-Ret is unclear. If no OCR information is included, how can the target image be retrieved, especially for answers that require precise text content? From my perspective, a pooling-based representation feature does not offer sufficient information.
>
> Thank you for raising this concern. Recent vision-language models (VLMs), such as the MiniCPM-V and Qwen2-VL series, have demonstrated a strong ability to understand documents directly from images, without requiring OCR information. Our proposed VisRAG builds upon MiniCPM-V and further validates this capability through extensive experiments on document RAG tasks.
>
> As shown in Table 8 of our paper’s appendix, we provide a case study to highlight this advantage. The user’s question is: “*On which day is Club Jetty closed?*” The relevant document, displayed on the right, includes the keyword “Club Jetty” written in a decorative font. OCR-based methods incorrectly recognize “Club Jetty” as “CLUBIETTY,” resulting in inaccurate text-based retrieval, as shown on the left. In contrast, our VLM-based end-to-end solution successfully identifies and comprehends the text, demonstrating its superior robustness to font variations.
>
> This example illustrates that relying on OCR can lead to retrieval errors due to text misrecognition, whereas our VisRAG leverages the inherent strengths of VLMs to achieve more accurate and reliable document understanding.
>
> To further address your concerns on VisRAG-Ret's ability in retrieving text-heavy documents, we conduct additional experiments on the text retrieval benchmark BEIR [1]. Specifically, we render document texts into screenshots and compare VisRAG-Ret’s performance against MiniCPM (OCR), the text-based retrieval model that directly processes raw document texts (we use “(OCR)” to maintain naming consistency). Both models are trained on the same amount of data under the in-domain setting (Table 2\(c) in our paper). Due to computational constraints, we evaluate them on three small-scale datasets from BEIR.
>
> The retrieval performance in NDCG@10 (the official metric of BEIR) is presented below and has been updated in our paper **(Table 13, Appendix I)**:
>
> \begin{array}{l | l | l | l}
> \hline
> \textbf{Model} & \textbf{SciFact} & \textbf{NFCorpus} & \textbf{Scidocs} \newline
> \hline
>  \text{MiniCPM (OCR)} & 61.04 & 14.12 & 13.01 \newline
>  \text{VisRAG-Ret} & 62.47 & 27.02 & 16.25 \newline
> \hline
> \end{array}
>
> As shown, by **utilizing only the rendered screenshots, VisRAG-Ret significantly outperforms MiniCPM (OCR) on documents containing only textual information**. This result highlights that VisRAG-Ret’s pooling-based representation effectively captures textual details and is well-suited for text-heavy document retrieval.

---

> ### Author Response · Authors · 2024-11-19
>
> > Weakness 2. The experimental setup is not sufficiently convincing, indicating that the VQA task alone cannot fully demonstrate the method's effectiveness in the retrieval task. In other words, employing such a complex pipeline to address the VQA task is akin to using a sledgehammer to crack a nut.
>
> Thank you for your comments. As we pointed out above, our work is not to address the VQA task. Instead, we are focusing on a challenging **multi-modal document RAG** task, which requires the retrieval of the relevant document(s) given user query. Thus our task is sufficient to demonstrate our method's effectiveness in retrieval, as retrieval is the bottleneck of our pipeline. We respectfully disagree with the characterization of our approach as “using a sledgehammer to crack a nut” as we remain a simple VLM-based RAG pipeline to address the document RAG task.

---

> ### Author Response · Authors · 2024-11-19
>
> > Weakness 3. The methods discussed are insufficient. To the best of my knowledge, there are several document-specific MLLMs, such as mlug-docowl, ureader, and text monkey, among others. While these works may not be directly related to RAG, they should at least be mentioned and analysized.
>
> Thank you for raising your concern regarding our method and we appreciate the opportunity to clarify our contributions and methodology. Our main contribution, VisRAG, is a VLM-based RAG **framework**, and is **model-agnostic**. This means that VisRAG can incorporate **any document-understanding VLM** as its backbone, allowing flexibility and adaptability to various VLM architectures. To address your concern, we 1) add the review of these document-specific VLMs in our Related Work section, and 2) provide a performance comparison on the DocVQA dataset [2] for the models you mentioned (DocOwl [3], UReader [4], and TextMonkey [5]) alongside MiniCPM-V series models in the table below. Please note that these results are from a VQA task, not the RAG task defined in our paper. Therefore, they cannot be directly compared to the results presented in our paper.
>
> |Model|Size|ANLS on DocVQA (Test)|
> |----|----|---|
> |DocOwl [3]|7B|62.2|
> |UReader [4] |7B|65.4|
> |TextMonkey [5] |9B|73.0|
> |DocOwl 1.5 [6] |8B|82.2|
> |DocOwl 2 [7] |8B|80.7|
> |MiniCPM-V 2.0 [8]|3B|71.9|
> |MiniCPM-V 2.6 [9]|8B|90.8|hl
>
> (Note: The performance of DocOwl, UReader, TextMonkey is from the DocOwl 2 paper [7]. Results for MiniCPM-V 2.0 and 2.6 are sourced from their respective Hugging Face README pages [8,9].)
>
> We select MiniCPM-V 2.0 and MiniCPM-V 2.6 as the backbone models for our implementation based on the following considerations:
>
> - MiniCPM-V 2.0, despite its smaller size (3B), demonstrates superior document understanding compared to DocOwl (7B), UReader (7B), and achieves comparable performance to TextMonkey (9B). This makes it an ideal choice for retriever tasks that involve encoding an entire corpus. Additionally, its lightweight architecture makes it suitable for generating answers efficiently.
> - MiniCPM-V 2.6 represents a state-of-the-art VLM, surpassing DocOwl 1/1.5/2, UReader, TextMonkey, and MiniCPM-V 2.0 in performance. Furthermore, its capability to accept multiple images as input makes it effective as the generator in our framework, synthesizing answers based on top-k retrieved documents.
>
> While we recognize the value of exploring other document-specific VLMs, our primary focus in this paper is on proposing a novel RAG framework for multi-modal documents, not on benchmarking different VLMs as backbones. Due to computational constraints, we opt for a targeted evaluation with strong-performing MiniCPM models. A comprehensive analysis of how different VLMs perform as VisRAG backbones lies beyond the scope of our current study.

---

> ### Author Response · Authors · 2024-11-19
>
> > Weakness 4. In my view, the comparison is unfair, as both Minicpm and Siglip can not only perform VQA tasks but also handle other tasks related to document image understanding and content perception.
>
> We acknowledge that VLMs such as MiniCPM and SigLIP are versatile and capable of handling a wide range of tasks, including those related to document image understanding and content perception. However, the focus of our study is **not on evaluating their general task performance** but on **adapting VLMs specifically for multi-modal document RAG**. While MiniCPM and SigLIP may excel in other domains, benchmarking their performance across diverse tasks lies beyond the scope of this work. Instead, we emphasize the importance of understanding how these models perform in a RAG setting, which is the central contribution of our research.
>
> We appreciate your thoughtful feedback and is willing to provide further clarifications if needed.
>
> References
>
> [1] Thakur et al. BEIR: A Heterogenous Benchmark for Zero-shot Evaluation of Information Retrieval Models. NeurIPS 2021.
>
> [2] Mathew et al. DocVQA: A dataset for VQA on document images. WACV 2021.
>
> [3] Ye et al. mPLUG-DocOwl: Modularized Multimodal Large Language Model for Document Understanding. Arxiv 2023.
>
> [4] Ye et al. Ureader: Universal ocrfree visually-situated language understanding with multimodal large language model. EMNLP 2023 (Findings).
>
> [5] Liu et al. Textmonkey: An ocr-free large multimodal model for understanding document. Arxiv 2024.
>
> [6] Hu et al. mPLUG-DocOwl 1.5: Unified Structure Learning for OCR-free Document Understanding. EMNLP 2024.
>
> [7] Hu et al. mPLUG-DocOwl2: High-resolution Compressing for OCR-free Multi-page Document Understanding. Arxiv 2024.
>
> [8] OpenBMB. openbmb/minicpm-v-2, 2024a. https://huggingface.co/openbmb/MiniCPM-V-2.
>
> [9] OpenBMB. openbmb/minicpm-v-2 6, 2024b. https://huggingface.co/openbmb/MiniCPM-V-2_6.

---

> ### Author Response · Authors · 2024-11-24
> **A gentle reminder**
>
> Dear Reviewer kCGC,
>
> Thank you for your time and efforts again in reviewing our paper. We kindly remind you that most of the discussion period has passed. We have carefully addressed the comments and suggestions you provided in our submitted rebuttal, and we hope our responses clarify any concerns or questions.
>
> If you have any additional feedback or require further clarification, please feel free to reach out. Thank you very much for your attention.
>
> Best regards,
>
> Authors

---

> ### Comment · Reviewer_kCGC · 2024-11-29
>
> Thanks for the detailed rebuttal, where most of my concerns have been solved properly. I thus would like to raise the score to 6.

---

> > ### Author Response · Authors · 2024-11-30
> >
> > Thank you for carefully reviewing our rebuttal and raising the score. We truly appreciate your time and feedback throughout the review process.

---

### Official Review · Reviewer_ZBwP · 2024-11-03

**Soundness:** 3
**Presentation:** 4
**Contribution:** 3
**Rating:** 6
**Confidence:** 4

**Summary:**

The paper focus on the research of vision-language model (VLM)-based RAG
Pipeline and propose a novel method named VisRAG. It  broadens the external data processed by RAG from text to image. To achieve the goal, the authors introduce two vision-optimized modules: VisRAG-Ret and VisRAG-Gen. This VisRAG-Ret directly utilizes the images without extracted textual content and establish  the map between the query and the documents in the latent space. The VisRAG-Gen leverage the help of the existing LLMs and VLMs for generating the answers. The image processing technology is integrated to realize the efficient generation of multiple inputs. Moreover, a new dataset is built combining a vision question answering (VQA) dataset and synthetic data.

**Strengths:**

In this paper, the authors propose an image-based RAG method named VisRAG, and construct a new dataset to verify the relevant effects. This paper does a lot of comparative experiments to verify its innovation. A large number of state-of-the-art methods have been combined and proven to be effective.

**Weaknesses:**

The method in this paper is mainly based on the retrieval of image data, which is basically consistent with the text retrieval framework, and lacks innovation. In addition, similar to the comparative model Colpali's innovation, the VisRAG is fine-tuned with a new dataset and new VLM which is not sufficient to express its innovation in multimodality area.

**Questions:**

1. In the comparison process, Colpali and Vis RAG-Ret should be fine-tuned based on the same data, rather than using the pre-trained model to compare directly. Please add more comparing results.
2.The article needs to provide more comparison results with the current best models, such as GPT o1, Qwen and other leading models that can directly input image for VQA task.

---

> ### Author Response · Authors · 2024-11-19
>
> Thank you for your detailed review. Please find our point-by-point response to your comments/questions below.
>
> > Weakness. The method in this paper is mainly based on the retrieval of image data, which is basically consistent with the text retrieval framework, and lacks innovation. In addition, similar to the comparative model Colpali's innovation, the VisRAG is fine-tuned with a new dataset and new VLM which is not sufficient to express its innovation in multimodality area.
>
> Thank you for your thoughtful comments. We respectfully argue that our work presents a novel contribution to the field of RAG. Specifically:
>
> 1. Introduction of a novel RAG paradigm: VisRAG introduces, as far as we know, the first fully vision-language model (VLM)-based RAG pipeline. Unlike traditional methods that rely on document parsing, VisRAG handles multi-modal documents in a more end-to-end way, completely removing these extra steps. **This represents a significant paradigm shift from traditional approaches.**
> 2. Development of a multi-modality document RAG dataset: We propose a new multi-modal document RAG dataset derived from VQA datasets and synthetic data. By employing filtering techniques, this dataset is designed to effectively train and evaluate VisRAG and baselines. This contribution addresses a significant gap, **providing a benchmark that can facilitate future research in multi-modal document RAG tasks**.
> 3. Demonstration of better performance with a simpler pipeline: Through extensive experiments, we demonstrate that our simple, streamlined approach VisRAG surpasses methods that involve parsing texts from multi-modal documents—solutions widely adopted and validated in industrial frameworks like LlamaIndex. For example, in Table 2 and Table 5, we show that VisRAG performs better than pipelines using extracted texts from OCR tools with our carefully designed post-processing strategy or from MiniCPM (Captioner), a model trained on GPT-4V-generated data for transcribing document images into texts. We also show that VisRAG can surpass SOTA text embedding models with extracted texts by training on much less out-of-domain data (Table 2, Figure 4). VisRAG also works well with text-heavy documents (Figure 5). Our work **breaks the inherent belief that a good document RAG pipeline should rely on a manually tuned document parsing module and a strong text embedding model.** Instead, VisRAG shows that a simpler, more direct approach can work better.
>
> While our approach is simple, we contend that simplicity should not be equated with a lack of novelty. VisRAG’s novel design highlights how leveraging VLMs directly, without intermediate parsing, can redefine the boundaries of RAG systems. This new architecture not only enhances effectiveness but also provides a fresh perspective for tackling multi-modal document tasks.
>
> In response to your concerns regarding ColPali, first, we would like to emphasize that according to ICLR 2025's [Reviewer Guide](https://iclr.cc/Conferences/2025/ReviewerGuide), ColPali is **contemporaneous work**, as it is a recent pre-print and is also [a submission to ICLR 2025](https://openreview.net/forum?id=ogjBpZ8uSi). Despite this, VisRAG significantly differs from ColPali in several critical aspects:
>
> 1.	Data construction methodology: Similar to ColPali, we construct data from VQA datasets, but we identify that there are a large portion of queries from VQA datasets are context-depedent and not suitable for retrieval(see Section 3.3). To address this, we propose an additional filtering step leveraging an LLM to exclude these unsuitable queries. This refinement ensures that the dataset used for VisRAG is better aligned with RAG tasks, making our evaluation **more reliable and representative of real-world use cases**.
> 2.	Scope and pipeline design: VisRAG is the **first full vision-based RAG pipeline**, whereas ColPali functions solely as a retriever. Our work integrates VLMs throughout the entire RAG pipeline, setting it apart in its comprehensive approach. Furthermore, we conduct extensive studies to show that the VisRAG pipeline can outperform TextRAG pipelines on real-world documents including text-heavy ones (Figure 5) and demonstrates strong generalization ability. This innovation represents a significant advancement in multi-modal document RAG.
> 3.	Retrieval efficiency: The retriever component of VisRAG, VisRAG-Ret, employs a **single-vector embedding model**, making it more efficient than ColPali’s retriever. Despite this simplicity, VisRAG-Ret demonstrates on-par performance compared to ColPali in fair evaluations using the same training data (See below). This highlights the effectiveness of our design.

---

> ### Author Response · Authors · 2024-11-19
>
> > Question 1. In the comparison process, Colpali and VisRAG-Ret should be fine-tuned based on the same data, rather than using the pre-trained model to compare directly. Please add more comparing results.
>
> Thank you for highlighting this concern. To address it, we fine-tune ColPali on the same dataset used for VisRAG-Ret. The updated results are shown below and have been incorporated into our paper **(Table 2)**.
>
> \begin{array}{l | l | l | l | l | l | l | l | l }
> \hline
> \textbf{Setting} & \textbf{Model} & \textbf{ArxivQA} & \textbf{ChartQA} & \textbf{DocVQA} & \textbf{InfoVQA} & \textbf{PlotQA} & \textbf{SlideVQA} & \textbf{Average} \newline
> \hline
> \textbf{In-domain} & \text{ColPali} & 64.61 & 59.18 & 82.17 & 79.19 & 33.49 & 93.08 & 68.62 \newline
> & \text{VisRAG-Ret} & 67.00 & 59.34 & 77.65 & 84.05 & 40.26 & 91.71 & 70.00 \newline
> \hline
> \end{array}
>
> The results indicate that VisRAG-Ret achieves comparable or superior performance to ColPali across the evaluated tasks. An important distinction lies in their efficiency: ColPali is a multi-vector model that generates 1,030 vectors (each 128-dimensional) for every document, while VisRAG-Ret produces a single 2,304-dimensional vector per document. This difference results in VisRAG-Ret consuming **only 1.7%** of the memory required by ColPali, making it significantly more memory-efficient and scalable.

---

> ### Author Response · Authors · 2024-11-19
>
> > Question 2. The article needs to provide more comparison results with the current best models, such as GPT o1, Qwen and other leading models that can directly input image for VQA task.
>
> Thank you for your valuable suggestions. Regarding your comment about including results for OpenAI o1, we would like to clarify that the current API of OpenAI o1 does not support image input (See [o1's documentation](https://platform.openai.com/docs/guides/reasoning/advice-on-prompting#beta-limitations)). Therefore, we are unable to include its results in our comparison.
>
> To address your concern, we have added the results of the Qwen2-VL-7B-Instruct model, a SOTA VLM capable of handling multi-image inputs for VQA tasks [1]. The average generation performance of Qwen2-VL-7B-Instruct is presented in the table below and in the updated paper **(Table 11, Appendix G)**:
>
> \begin{array}{l | l | l}
> \hline
> \textbf{Model/Method} & \textbf{Input} & \textbf{Average Accuracy (percent)} \newline
> \hline
>  & \text{top-1} & 53.48 \ (77.8\ \text{percent}) \newline
> \textbf{Qwen2-VL} & \text{top-2} & 54.86 \ (79.7\ \text{percent}) \newline
> \text{(Multi-image VLM)}    & \text{top-3} & 54.79 \ (79.5\ \text{percent}) \newline
>                      & \text{Oracle} & 68.91 \ (100\ \text{percent}) \newline
> \hline
> \end{array}
>
> The results show that Qwen2-VL exhibits superior document understanding capabilities, achieving ~5% higher performance than MiniCPM-V 2.6. This improvement is observed both in the oracle setting, where the model has access to the golden document(s), and in practical scenarios, where the model processes the top-1 to top-3 retrieved documents from VisRAG-Ret.
>
> References
>
> [1] Wang et al. Qwen2-VL: Enhancing Vision-Language Model's Perception of the World at Any Resolution. Arxiv 2024.

---

> ### Author Response · Authors · 2024-11-24
> **A gentle reminder**
>
> Dear Reviewer ZBwP,
>
> Thank you for your time and efforts again in reviewing our paper. We kindly remind you that most of the discussion period has passed. We have carefully addressed the comments and suggestions you provided in our submitted rebuttal, and we hope our responses clarify any concerns or questions.
>
> If you have any additional feedback or require further clarification, please feel free to reach out. Thank you very much for your attention.
>
> Best regards,
>
> Authors

---

> ### Author Response · Authors · 2024-12-02
>
> Dear Reviewer ZBwP,
>
> Thank you once again for taking the time to review our submission. As the discussion period deadline approaches, we would like to kindly follow up to see if you might have had a chance to review our rebuttal. We are happy to address any additional questions or concerns you may have.
>
> Thank you for your time and consideration.
>
> Best regards,
>
> Authors

---

### Official Review · Reviewer_gQ1V · 2024-11-03

**Soundness:** 3
**Presentation:** 3
**Contribution:** 2
**Rating:** 6
**Confidence:** 3

**Summary:**

The paper introduces VisRAG to address the limitations of text-only RAG systems by incorporating vision-language models (VLMs). VisRAG enables the processing of documents as images. By training the VisRAG retriever with open-source and synthetic data and exploring various generation methods, experimental results show VisRAG's superiority in both retrieval and generation tasks. The authors also plan to make their code and data publicly available.

**Strengths:**

1. Improve text-only RAG by establishing a vision-language model (VLM)-based RAG pipeline.
2. The experiment is thorough under the designed scenario.
3. The authors have promised to make the data and code open source.

**Weaknesses:**

1. Lack of novelty: For retrieval, it uses the dual-encoder-style and just changes them to VLM embeddings. The position-weighted mean pooling is also off-the-shelf. For generation, they only try a few straightforward strategies.
2. Lack of design for document understanding: There is a lack of design considerations specifically for document understanding. The proposed method is not tailored for document understanding scenarios. Alternatively, why not validate the method on non-document (common image) datasets?
3. Confusion regarding the application: Why does a person always need the assistance of document images when asking general questions? For example, in Line 884, Why would a person directly ask a question like 'Where were the two magnesium containing papers made at?' without reference to any document? If he already has this document as a reference, why does he still need to do a retrieval process at first?
4. Why only experiment with text-only or image-only features, not image+text(+layout)?

**Questions:**

See Weaknesses.

---

> ### Author Response · Authors · 2024-11-19
>
> Thank you for your detailed review. Please find our point-by-point response to your comments/questions below.
>
> > Weakness 1. Lack of novelty: For retrieval, it uses the dual-encoder-style and just changes them to VLM embeddings. The position-weighted mean pooling is also off-the-shelf. For generation, they only try a few straightforward strategies.
>
> Thank you for your thoughtful comments. We respectfully argue that our work presents a novel contribution to the field of RAG. Specifically:
>
> 1. Introduction of a novel RAG paradigm: VisRAG introduces, as far as we know, the first fully vision-language model (VLM)-based RAG pipeline. Unlike traditional methods that rely on document parsing, VisRAG handles multi-modal documents in a more end-to-end way, **completely removing these extra steps**. **This represents a significant paradigm shift from traditional approaches.**
> 2. Development of a multi-modality document RAG dataset: We propose a new multi-modal document RAG dataset derived from VQA datasets and synthetic data. By employing filtering techniques, this dataset is designed to effectively train and evaluate VisRAG and baselines. This contribution addresses a significant gap, **providing a benchmark that can facilitate future research in multi-modal document RAG tasks**.
> 3. Demonstration of better performance with a simpler pipeline: Through extensive experiments, we demonstrate that our simple, streamlined approach VisRAG surpasses methods that involve parsing texts from multi-modal documents—solutions widely adopted and validated in industrial frameworks like LlamaIndex. For example, in Table 2 and Table 5, we show that VisRAG performs better than pipelines using extracted texts from OCR tools with our carefully designed post-processing strategy or from MiniCPM (Captioner), a model trained on GPT-4V-generated data for transcribing document images into texts. We also show that VisRAG can surpass SOTA text embedding models with extracted texts by training on much less out-of-domain data (Table 2, Figure 4). VisRAG also works well with text-heavy documents (Figure 5). Our work **breaks the inherent belief that a good document RAG pipeline should rely on a manually tuned document parsing module and a strong text embedding model.** Instead, VisRAG shows that a simpler, more direct approach can work better.
>
> While our approach is simple, we contend that **simplicity should not be equated with a lack of novelty**. VisRAG’s novel design highlights how leveraging VLMs directly, without intermediate parsing, can redefine the boundaries of RAG systems. This new architecture not only enhances effectiveness but also provides a fresh perspective for tackling multi-modal document tasks.

---

> ### Author Response · Authors · 2024-11-19
>
> > Weakness 2. Lack of design for document understanding: There is a lack of design considerations specifically for document understanding. The proposed method is not tailored for document understanding scenarios. Alternatively, why not validate the method on non-document (common image) datasets?
>
> While VisRAG’s focus is to perform RAG on multi-modal documents, we acknowledge that VisRAG does not include designs specifically tailored for document understanding. The building blocks of VisRAG—state-of-the-art VLMs—already demonstrate strong document understanding capabilities.
>
> For example, the open-source VLMs used in our paper, MiniCPM-V 2.0 and 2.6, achieve impressive results on document understanding benchmarks including DocVQA [1], TextVQA [2], and OCRBench [3], surpassing VLMs that are designed explicitly for document understanding, such as TextMonkey [4] (see performance metrics detailed in the [GitHub README](https://github.com/OpenBMB/MiniCPM-V) and the technical report [5] (Table 5) of MiniCPM-V). Other leading VLMs including Qwen2-VL [6] and GPT-4o [7] have also demonstrated strong document understanding capabilities (refer to Qwen2-VL’s paper [6] for GPT-4o’s results).
>
> Given the robust document understanding capabilities of modern VLMs, adding document-specific designs, in our view, is unnecessary, and **could introduce inductive biases, potentially limiting VisRAG’s generalization**. Additionally, our approach allows any SOTA VLM to be seamlessly integrated into the pipeline. Our experimental results also show that VisRAG, without explicit document-oriented designs, consistently outperforms baseline methods.
>
> Although VisRAG(-Ret) is not specifically designed for document understanding, it is trained on multi-modal document retrieval data, unlike its vision encoder, SigLIP, which is trained with a focus on general image-text alignment. Like the reviewer, we are also curious about the extent to which VisRAG-Ret retains image retrieval capabilities. We evaluated it on the MS COCO image retrieval task [8], where the model retrieves everyday images based on their captions. VisRAG-Ret achieves a Recall@5 of 66.2, compared to 75.7 for the original SigLIP encoder. This demonstrates that, despite being trained exclusively for document retrieval, VisRAG-Ret retains 87.5% of the original vision encoder’s image retrieval performance. Additionally, it excels in document retrieval, achieving an MRR@10 of 70.0 on our dataset, significantly outperforming SigLIP, which achieves only 49.41. We agree that developing a retriever capable of handling both multi-modal document retrieval and image retrieval tasks is an intriguing direction for future work.

---

> ### Author Response · Authors · 2024-11-19
>
> > Weakness 3. Confusion regarding the application: Why does a person always need the assistance of document images when asking general questions? For example, in Line 884, Why would a person directly ask a question like 'Where were the two magnesium containing papers made at?' without reference to any document? If he already has this document as a reference, why does he still need to do a retrieval process at first?
>
> Thank you for highlighting this issue. As outlined in Section 3.3, part of our dataset is constructed from VQA datasets, where the queries are often generated by workers with access to the corresponding documents. This leads to a high proportion of context-dependent queries, which become nonsensical when viewed independently. To mitigate this, we implement a filtering process using an LLM prompted with in-context human-annotated examples, to identify and exclude such context-dependent queries. Despite these efforts, borderline cases—queries that exhibit subtle context dependence—pose a classification challenge. Both human annotators and LLMs can occasionally misclassify such cases, as perfect separation of all context-dependent queries is inherently complex.
>
> The specific query you mentioned, “Where were the two magnesium-containing papers made at?”, illustrates this challenge. Its ambiguity regarding “the two magnesium containing papers” inherently assumes prior document knowledge, making it unsuitable for standalone retrieval. We acknowledge that such a query should have been flagged as context-dependent during the filtering process and excluded from the dataset.
>
> However, we emphasize that the potential presence of such isolated noise is minimal and does not significantly affect the quality or utility of the dataset. Our experiments demonstrate that VisRAG consistently outperforms baseline models, underscoring the robustness of our dataset in effectively evaluating retrieval tasks.
>
> We appreciate your observation and will incorporate additional steps in future work to refine the filtering process and reduce the occurrence of such queries.

---

> ### Author Response · Authors · 2024-11-19
>
> > Weakness 4. Why only experiment with text-only or image-only features, not image+text(+layout)?
>
> Thank you for raising this concern regarding the scope of our experiments. We intentionally choose text-only baselines such as OCR-based TextRAG and TextRAG with VLM-transcribed texts because these represent the current standard practices in industrial RAG frameworks. For instance, [LlamaParse](https://docs.llamaindex.ai/en/stable/llama_cloud/llama_parse/) from [LlamaIndex](https://www.llamaindex.ai/) employs OCR-based methods and VLMs to convert document images into textual representations, forming the foundation of its pipeline (refer to [LlamaParse documentation](https://docs.cloud.llamaindex.ai/llamaparse/features/parsing_options) and [premium features](https://www.llamaindex.ai/blog/introducing-llamaparse-premium)). In line with these industrial trends, we utilize the open-source OCR tool [PPOCR](https://github.com/PaddlePaddle/PaddleOCR/blob/main/README_en.md) to establish a reliable OCR-based baseline and employ our trained MiniCPM-V 2.0 captioner for VLM-based transcription. These approaches aim to closely mimic real-world pipelines. Additionally, we adopt SigLIP [9], the vision encoder of VisRAG’s backbone model MiniCPM-V, as the image-only baseline to directly encode document images (screenshots/scans) to serve as an ablation study of our VisRAG architecture.
>
> Regarding the inclusion of an image+text+layout baseline, we acknowledge its potential but emphasize the technical challenges involved. Currently, **there is no established method for effectively parsing document images into interleaved text and figures and then embedding them with layout information**. Existing approaches, such as UniIR [10] and MARVEL [11], focus on paired (image, text) datasets as the retrieval target, as highlighted in our Introduction and Related Work sections. However, extracting such pairs from real-world documents with diverse layouts is infeasible. Thus these methods cannot be applied to our multi-modal document RAG scenario. We recognize that building a RAG pipeline capable of processing extracted images and texts with layout information preserved is an interesting research direction. However, it is beyond the scope of our current work. We regard this as an area for future research.
>
> Though it is infeasible now to build an image+text+layout baseline, to address the reviewer’s concern, we build an image+text baseline where we use reciprocal rank fusion (RRF) [12] to combine the retrieval result from SigLIP, which directly encodes the document image (screenshots/scans), and the text retriever MiniCPM (OCR), which encodes the text extracted by an OCR module. This baseline considers both the text and the image modality. We conduct experiments for this new baseline under both out-of-domain and in-domain settings, consistent with our paper. We present the results below and in our updated paper **(Table 10, Appendix G)**.
>
> \begin{array}{l | l | l}
> \hline
> \textbf{Setting} & \textbf{Model} & \textbf{Average MRR@10} \newline
> \hline
> & \text{MiniCPM (OCR)} & 55.53 \newline
> \textbf{Out-of-domain} & \text{SigLIP} & 52.46 \newline
> & \text{MiniCPM (OCR) + SigLIP (RRF)} & 57.21 \newline
> \hline
> & \text{MiniCPM (OCR)} & 64.64 \newline
> \textbf{In-domain} & \text{SigLIP} & 63.37 \newline
> & \text{MiniCPM (OCR) + SigLIP (RRF)} & 67.06 \newline
> \hline
> \end{array}
>
> The results indicate that fusing text and image modalities provides a meaningful performance boost over individual modality baselines. However, this approach still falls short of the performance achieved by our VisRAG-Ret model (63.96 for out-of-domain, 70.00 for in-domain). This underscores the superior capability of VisRAG-Ret in understanding both modalities within a unified architecture.
>
> References
>
> [1] Mathew et al. DocVQA: A dataset for VQA on document images. WACV 2021.
>
> [2] Singh et al. Towards VQA models that can read. CVPR 2019.
>
> [3] Liu et al. On the hidden mystery of OCR in large multimodal models. Arxiv 2023.
>
> [4] Liu et al. Textmonkey: An ocr-free large multimodal model for understanding document.
>
> [5] Yao et al. Minicpm-v: A gpt-4v level mllm on your phone. Arxiv 2024.
>
> [6] Wang et al. Qwen2-VL: Enhancing Vision-Language Model's Perception of the World at Any Resolution. Arxiv 2024.
>
> [7] OpenAI. Hello, gpt-4o — openai, 2024. https://openai.com/index/hello-gpt-4o/.
>
> [8] Lin et al. Microsoft coco: Common objects in context. ECCV 2014.
>
> [9] Zhai et al. Sigmoid Loss for Language Image Pre-Training. ICCV 2023.
>
> [10] Wei et al. Uniir: Training and benchmarking universal multimodal information retrievers. Arxiv 2023.
>
> [11] Zhou et al. MARVEL: unlocking the multi-modal capability of dense retrieval via visual module plugin. ACL 2024.
>
> [12] Cormack et al. Reciprocal rank fusion outperforms condorcet and individual rank learning methods. SIGIR 2009.

---

> ### Author Response · Authors · 2024-11-24
> **A gentle reminder**
>
> Dear Reviewer gQ1V,
>
> Thank you for your time and efforts again in reviewing our paper. We kindly remind you that most of the discussion period has passed. We have carefully addressed the comments and suggestions you provided in our submitted rebuttal, and we hope our responses clarify any concerns or questions.
>
> If you have any additional feedback or require further clarification, please feel free to reach out. Thank you very much for your attention.
>
> Best regards,
>
> Authors

---

> > ### Comment · Reviewer_gQ1V · 2024-11-27
> > **Response to Authors**
> >
> > Thanks for the authors' rebuttal. However, I still think that ''Weakness 3. Confusion regarding the application'' is an unresolved issue that cannot be totally ignored. I will keep my current rating.

---

> ### Author Response · Authors · 2024-11-30
>
> Thank you for raising your concern again regarding the application of VisRAG. In our previous rebuttal, we clarified that the query you mentioned, “Where were the two magnesium-containing papers made at?”, should be treated as a context-dependent query. Such queries require the document to be explicitly presented for comprehension and are therefore unsuitable for retrieval tasks. We acknowledged that this specific example was incorrectly labeled by our annotators, which may have resulted in a small number of context-dependent queries not being filtered out by the LLM. However, we argued that the presence of such cases is minimal and does not significantly affect the overall quality of our dataset.
>
> **We would like to emphasize that the presence of this noise in our dataset does not diminish the applications of VisRAG.** VisRAG is effective for scenarios where users have a knowledge base comprising one or more documents (here, a “document” refers to a real-world entity, which could span multiple pages, differing from the definition used in our paper, where a “document” means a page) and need to query specific information contained within them. For example, a student could index all their textbooks (including illustrations) using VisRAG and ask questions to aid their learning process. Similarly, a company could implement a question-answering system powered by VisRAG by indexing documents such as regulatory guidelines or machine part manuals, enabling staff to query them. In practical applications, **users are highly unlikely to pose ambiguous queries** like “Where were the two magnesium-containing papers made at?” without specifying the context, as they would naturally recognize such queries as incomplete or nonsensical. If such queries do arise, a robust system could incorporate mechanisms to ask clarifying follow-up questions. However, designing such interaction models is outside the scope of our current work.
>
> We've provided a study on two cases of VisRAG in Table 8 and Table 9 of Appendix F in our paper. To further address your concern, we provide three potential use cases of VisRAG in this [anonymized URL](https://github.com/anonymous-account-for-iclr-2025/rebuttal/blob/42518d3be8e06ccbc462493bd516ec2cde5c5062/visrag_application.pdf) with a manual and a datasheet sourced from the web as the knowledge bases. These examples showcase how VisRAG effectively handles personalized information needs when applied to knowledge bases that are distinct from our training and evaluation datasets.
>
> We sincerely appreciate your thoughtful feedback and are happy to provide further clarifications if needed.

---

> > ### Comment · Reviewer_gQ1V · 2024-11-30
> > **Response to Authors**
> >
> > Thanks again for the authors' rebuttal.
> >
> > Actually, I still have concerns about Weakness 1 and 2. However, the authors have made sufficient efforts and the work has achieved certain results. Therefore, I will raise my score and I sincerely hope that the authors can refine the proposed dataset in the camera-ready version of this paper.

---

> > > ### Author Response · Authors · 2024-12-02
> > >
> > > Thank you for carefully reviewing our rebuttal and for raising your score. We sincerely appreciate your thoughtful feedback, which has been instrumental in improving our work.
> > >
> > > To address the concerns you highlighted, we plan to take the following steps in the next version of our paper:
> > >
> > > 1. Explicitly acknowledge the issue of potential noise in our dataset and discuss its implications.
> > > 2. Employ a larger LLM with reviewed human-annotated examples to effectively re-filter the evaluation data and minimize noise.
> > >
> > > Thank you again for your time and constructive suggestions.

---

### Official Review · Reviewer_A94Q · 2024-11-04

**Soundness:** 3
**Presentation:** 3
**Contribution:** 3
**Rating:** 6
**Confidence:** 4

**Summary:**

This paper proposes VisRAG, a multimodal retrieval-augmented generation pipeline. Different from traditional RAG methods, which rely on text parsing to retrieve information from visual content, this paper utilizes a VLM-based retriever and generator to navigate information relevant to the query and generate responses. The VisRAG design consists of two main stages: (1) Retrieval: Given a query, the VLM retrieves a set of relevant images from the dataset leveraging the cosine similarity between the query embedding and image embeddings; (2) Generation: the VLM uses a combination of retrieved images and the query to produce responses. VisRAG is compared with a series of baselines and show advantages in both retrieval and generation capabilities.

**Strengths:**

1. [Originality] The proposed method is promising. With the VLMs showing increasing capabilities in understanding images, encoding images with VLMs is a natural improvement from traditional text-based retrieval methods.
2. [Quality] The paper presents relatively comprehensive experiments, demonstrating advantages in both retrieval and generation capabilities of the proposed methods.
3. [Significance] The proposed method clearly outperforms various baselines.
4. [Clarity] The paper is generally well-written and easy to follow.

**Weaknesses:**

Major concerns:
1. In the paper, when retrieving relevant images, the authors adopt position-weighted mean pooling over the last-layer VLM hidden states. This choice seems heuristic-based. Although the paper points out that VLMs are with causal attention, potentially emphasizing weights on later tokens, after multiple VLM layers, we could expect information to propagate across most tokens. In fact, previous works often use the leading token or simply learn a linear layer from all tokens. How would the model perform with these designs? Would the proposed position-weighted mean pooling produce significantly better results?
2. When using multiple images to generate responses, one method proposed in the paper is image concatenation, which is performed horizontally. This could change the resolution of the final image. Would this affect VLM performance, particularly when handling a larger number of retrieved images (e.g., more than five)?
3. The paper does not report retrieval efficiency compared to text-based methods. Would using the VLM significantly increase retrieval time? How would the number of retrieved images affect retrieval time?

Minor comments:
1. When measuring the performance of generation, how is accuracy measured? To me, this is particularly unclear for synthetic data, where ground truth answers and queries are generated by GPT-4. Is an additional LLM used to evaluate the textual responses?

**Questions:**

1. Significance of position-weighted mean pooling: How would the model perform with these designs? Would the proposed position-weighted mean pooling produce significantly better results?
2. Effects of concatenated multiple images: Would this affect VLM performance, particularly when handling a larger number of retrieved images (e.g., more than five)?
3. Efficiency of VLM-based retrieval: Would using the VLM significantly increase retrieval time? How would the number of retrieved images affect retrieval time?
4. Clarity on accuracy measurement: How is accuracy measured? Is an additional LLM used to evaluate the textual responses?

---

> ### Author Response · Authors · 2024-11-19
>
> Thank you for your detailed review. Please find our point-by-point response to your comments/questions below.
>
> > Weakness 1. In the paper, when retrieving relevant images, the authors adopt position-weighted mean pooling over the last-layer VLM hidden states. This choice seems heuristic-based. Although the paper points out that VLMs are with causal attention, potentially emphasizing weights on later tokens, after multiple VLM layers, we could expect information to propagate across most tokens. In fact, previous works often use the leading token or simply learn a linear layer from all tokens. How would the model perform with these designs? Would the proposed position-weighted mean pooling produce significantly better results?
>
> > Question 1. Significance of position-weighted mean pooling: How would the model perform with these designs? Would the proposed position-weighted mean pooling produce significantly better results?
>
> Thank you for your insightful comments. Due to our limited computational budget and the architecture of VisRAG-Ret—a decoder-only model similar to GPT-Neo/GPT-J [1,2] as used in the SGPT paper [3]—we chose to adopt position-weighted mean pooling without conducting extensive comparative experiments with other pooling techniques. The SGPT paper [3] shows that position-weighted mean pooling performs better than other pooling techniques in text embedding tasks. Position-weighted mean pooling is a commonly used approach in text embedding models, supported by widely adopted toolkits such as [Sentence Transformers](https://www.sbert.net/) (See https://github.com/UKPLab/sentence-transformers/blob/master/sentence_transformers/models/Pooling.py).
>
> The primary contribution of our work is the introduction of the VisRAG framework, which integrates both the retriever and generator modules based on vision-language models (VLMs) capable of directly processing document page images. While we recognize the value of systematically optimizing each module, including experimenting with alternative pooling strategies, this lies beyond the scope of our current study and exceeds our computational resources. Our goal is to demonstrate the feasibility and promise of the VisRAG paradigm rather than achieve optimal performance for each module.
>
> Despite this, we observe that the position-weighted mean pooling still performs robustly, enabling VisRAG to significantly outperform traditional text-based pipelines. These results highlight VisRAG’s potential as an RAG framework for real-world documents. We leave further exploration into optimal designs for VisRAG-Ret including a more thorough evaluation of pooling strategies to future work.

---

> ### Author Response · Authors · 2024-11-19
>
> > Weakness 2. When using multiple images to generate responses, one method proposed in the paper is image concatenation, which is performed horizontally. This could change the resolution of the final image. Would this affect VLM performance, particularly when handling a larger number of retrieved images (e.g., more than five)?
>
> > Question 2. Effects of concatenated multiple images: Would this affect VLM performance, particularly when handling a larger number of retrieved images (e.g., more than five)?
>
> Thank you for highlighting this issue. Given our computational constraints, we conduct additional experiments on VisRAG-Gen using the top-6 and top-10 documents retrieved by VisRAG-Ret. We compare two models: MiniCPM-V 2.0, which can only accept a single image as input (using horizontal page concatenation), and MiniCPM-V 2.6, which supports multiple image inputs. Due to budget limitations, we do not conduct experiments on GPT-4o.
>
> The average generation performance is summarized in the table below, and we have updated our paper **(Table 11, Appendix G)** to include these detailed results.
>
> \begin{array}{l | l | l}
> \hline
> \textbf{Model/Method} & \textbf{Input} & \textbf{Average Accuracy (percent)} \newline
> \hline
> \textbf{Page Concatenation} & \text{top-6} & 24.83 \ (58.9\ \text{percent}) \newline
> \text{(Single-image VLM} & \text{top-10} & 23.87 \ (57.0\ \text{percent}) \newline
> \text{MiniCPM-V 2.0)} & \text{Oracle} & 40.60 \ (100\ \text{percent}) \newline
> \hline
> \textbf{MiniCPM-V 2.6} & \text{top-6} & 50.35 \ (76.8\ \text{percent}) \newline
>  \text{(Multi-image VLM)}  & \text{top-10} & 44.96 \ (68.8\ \text{percent}) \newline
>                      & \text{Oracle} & 65.32 \ (100\ \text{percent}) \newline
> \hline
> \end{array}
>
> The results reveal significant performance degradation when handling a larger number of retrieved pages, both for page concatenation (MiniCPM-V 2.0) and multi-page input (MiniCPM-V 2.6). Specifically, MiniCPM-V 2.0 drops from 34.44 (84.8%) for top-1 to 23.87 (57.0%) for top-10 when using page concatenation, and MiniCPM-V 2.6 decreases from 50.91 (77.9%) for top-1 to 44.96 (68.8%) for top-10 with multi-page input. Although both approaches show a decline, MiniCPM-V 2.6 demonstrates greater robustness to increasing context compared to MiniCPM-V 2.0.
>
> Our findings reveal the limitations of current VLMs in reasoning over multiple pages and extracting relevant information amidst noisy retrieved data. We believe that enhancing the ability of VisRAG-Gen to handle larger and noisier contexts is an essential direction for future research.

---

> ### Author Response · Authors · 2024-11-19
>
> > Weakness 3. The paper does not report retrieval efficiency compared to text-based methods. Would using the VLM significantly increase retrieval time? How would the number of retrieved images affect retrieval time?
>
> > Question 3. Would using the VLM significantly increase retrieval time? How would the number of retrieved images affect retrieval time?
>
> Thank you for raising your concern regarding retrieval efficiency. To address it, we conduct a comparison between VisRAG-Ret and MiniCPM (OCR), the baseline text-based retrieval model, in terms of both offline document processing latency and online query processing latency. The results are summarized in the table below.
>
> \begin{array}{l | lll | lll}
> \hline
>  & \textbf{Offline Latency} & \textbf{(ms)} & & \textbf{Online Latency} & \textbf{(ms)} & \newline
>  & \text{Parsing} & \text{Encoding} & \text{Total} & \text{Encoding} & \text{Search} & \text{Total} \newline
> \hline
> \text{MiniCPM (OCR)} & 284 & 28 & 312 & 28 & 26 & 54 \newline
> \text{VisRAG-Ret} & \text{--} & 121 & 121 & 28 & 26 & 54 \newline
> \hline
> \end{array}
>
> Our findings indicate that while VisRAG-Ret requires more time for document encoding compared to MiniCPM (OCR), it eliminates the time-intensive parsing step required by MiniCPM (OCR). This optimization leads to a 58% reduction in total document processing time for VisRAG-Ret. For online query processing, the latencies of VisRAG-Ret and MiniCPM (OCR) are nearly identical, as the queries consist solely of textual inputs.
>
> These results demonstrate that VisRAG-Ret achieves high efficiency while outperforming text-based retrievers in performance. We've updated our paper to include this experiment in **Appendix H**.
>
> You also raised a question about the impact of the number of retrieved images on retrieval time. In our implementation, we use the `torch.topk` function, which has a time complexity of $O(n + k)$ (See https://discuss.pytorch.org/t/whats-the-time-complexity-of-tensor-topk/117856/3), where $n$ is the total number of images in the corpus and $k$ is the number of retrieved images. Given that $k \ll n$ in our scenario, increasing $k$ has a negligible effect on retrieval time. Empirically, we observe no significant increase in retrieval time when $k$ grows from 0 to 20.

---

> ### Author Response · Authors · 2024-11-19
>
> > Weakness 4. When measuring the performance of generation, how is accuracy measured? To me, this is particularly unclear for synthetic data, where ground truth answers and queries are generated by GPT-4. Is an additional LLM used to evaluate the textual responses?
>
> > Question 4. Clarity on accuracy measurement: How is accuracy measured? Is an additional LLM used to evaluate the textual responses?
>
> In this paper, we focus solely on synthesizing additional **training** data, as described in Section 3.3 and Table 1. It is important to clarify that these synthetic data are **not used for evaluation purposes**. Our evaluations are conducted on datasets derived from established VQA benchmarks, including ArXivQA, ChartQA, MP-DocVQA, InfoVQA, PlotQA, and SlideVQA. These datasets provide concise ground-truth answers, enabling us to perform a (relaxed) exact match comparison.
>
> To evaluate model performance, we prompt VLMs to generate short-form answers (please refer to Table 7 for our prompts). The generated answers are then directly compared to the ground-truth answers, with a 5% allowed error margin for numeric responses. **We do not employ LLMs to evaluate textual responses.**
>
> We appreciate the reviewer’s feedback and have revised Section 3.3 to clarify the evaluation process.
>
> References
>
> [1] Andonian et al. GPT-NeoX: Large scale autoregressive language modeling in pytorch. Github 2021.
>
> [2] Wang et al. GPT-J-6B: A 6 Billion Parameter Autoregressive
> Language Model. Github 2021.
>
> [3] Niklas Muennighoff. Sgpt: Gpt sentence embeddings for semantic search. Arxiv 2022.

---

> ### Author Response · Authors · 2024-11-24
> **A gentle reminder**
>
> Dear Reviewer A94Q,
>
> Thank you for your time and efforts again in reviewing our paper. We kindly remind you that most of the discussion period has passed. We have carefully addressed the comments and suggestions you provided in our submitted rebuttal, and we hope our responses clarify any concerns or questions.
>
> If you have any additional feedback or require further clarification, please feel free to reach out. Thank you very much for your attention.
>
> Best regards,
>
> Authors

---

> ### Comment · Reviewer_A94Q · 2024-11-26
> **Thank you for the rebuttal**
>
> Thank you for the authors' rebuttal, which has resolved my previous concerns and clarify my misunderstandings. I will keep my current rating.

---

> > ### Author Response · Authors · 2024-11-26
> >
> > Thank you for carefully reviewing our rebuttal. We truly appreciate your time and feedback throughout the review process.

---

### Author Response · Authors · 2024-11-19
**General Rebuttal**

We sincerely thank all reviewers for their valuable feedback and constructive comments, which have helped us improve the quality of our work. In this rebuttal, we try to address each reviewer’s concerns and suggestions in detail in the individual responses. Below, we summarize the changes we made to our revised paper:

1. We have added **Appendix G** to present additional retrieval and generation results. This includes the retrieval performance of reciprocal rank fusion (RRF) between MiniCPM (OCR) and SigLIP (Table 10) and the generation performance when feeding the top-6 and top-10 retrieved documents to the model, as well as the performance of Qwen2-VL (Table 11).
2. We've added **Appendix H** to present the retrieval efficiency of VisRAG-Ret and MiniCPM (OCR) (Table 12).
3. We've added **Appendix I** to report the performance of VisRAG-Ret and MiniCPM (OCR) on text retrieval benchmarks.
4. The **Related Work** section has been updated to include a review of document-specific vision-language models (VLMs).
5. **Section 3.3 (Data Construction)** has been revised to explicitly state the datasets used for evaluation.
6. **Table 2, Section 4 (Experimental Methodology)** has been updated to reflect the results for ColPali trained on our dataset.
7. We identified and corrected problems in our previous implementation and paper, which affected the off-the-shelf performance of SigLIP **(Table 2)** and the generation performance of the page concatenation method on the SlideVQA dataset **(Table 3)**. The revised results now reflect the correct performance. Note that the average generation performance is correct in the previous version.

For detailed responses to your individual comments, please refer to the corresponding replies. Thank you again for your time and valuable insights!

---

### Meta-Review · Area_Chair_YWjP · 2024-12-19

**Metareview:**

This paper proposes VisRAG, a new multimodal retrieval augmented generation pipeline that use vision language models to retrieve relevant images based on a query and generate response for multi-modality documents. It consists two steps: retrieving relevant images using cosine similarity and generating answers by combining these images with the query. The extensive evaluations show its competitive performance on retrieval and generation capabilities.


The reviewers appreciate the presentation clarity of the proposed VisRag framework, and with comprehensive and promising results on various document related VQA tasks. Several questions and concerns are raised initially, including 1. the use of position-weighted mean pooling for retrieving relevant images method seems heuristic and its performance seems not clearly superior to other methods. There were also concerns raised about the technical novelty during the first round of review, particularly regarding its similarity to existing dual-encoder-style retrieval methods. However, with a detailed clarification on removing the document parsing preprocessing step, along with the additional merits of providing a new benchmark and strong experimental performance, all the reviewers were consistently positive about accepting the paper. With all these, I agree with the reviewers' unanimous recommendation for acceptance.

**Additional Comments On Reviewer Discussion:**

In the initial comment, there are several major concerning points are raised, for instance, 1. the heuristic method used and its unclear performance (summarized above as well) 2. Image concatenation and retrieval efficiency via concatenating multiple images horizontally may affect the VLM’s performance. 3. Novelty concerns and the practical application confusion. The authors explain the rationale to use position-weighted mean pooling over the last-lyaer VLM hidden state given the limitation of computation budget and the architecture of VisRAG-Vet. The authors also refer the choice of design to several existing works. For limitation of image concatenation, the authors provided new experimental results by comparing with MiniCPM-V2.0 and MiniCPM-V 2.6. The results reveal that reasoning over multiple pages and extracting relevant information is a challenge, which is a widely recognized difficult problem. Regarding the application confusion and the question about text-only and image-only experiments, the authors respond with a specific example and additional experiments. With the detailed response provided, no further questions arose from the reviewer.

---

### Decision · Program_Chairs · 2025-01-22

Accept (Poster)